# The importance of biofilm formation for cultivation of a Micrarchaeon and its interactions with its *Thermoplasmatales* host

Susanne Krause[1,14], Sabrina Gfrerer [1,2,14], Andriko von Kügelgen[3], Carsten Reuse[4,5], Nina Dombrowski [6], Laura Villanueva [6,7], Boyke Bunk [8], Cathrin Spröer[8], Thomas R. Neu [9], Ute Kuhlicke[9], Kerstin Schmidt-Hohagen[4,5], Karsten Hiller [4,5], Tanmay A. M. Bharat [3,10], Reinhard Rachel [11], Anja Spang [6,12] & Johannes Gescher[1,2,13✉]

Micrarchaeota is a distinctive lineage assigned to the DPANN archaea, which includes poorly characterised microorganisms with reduced genomes that likely depend on interactions with hosts for growth and survival. Here, we report the enrichment of a stable co-culture of a member of the Micrarchaeota (*Ca.* Micrarchaeum harzensis) together with its *Thermoplasmatales* host (*Ca.* Scheffleriplasma hospitalis), as well as the isolation of the latter. We show that symbiont-host interactions depend on biofilm formation as evidenced by growth experiments, comparative transcriptomic analyses and electron microscopy. In addition, genomic, metabolomic, extracellular polymeric substances and lipid content analyses indicate that the Micrarchaeon symbiont relies on the acquisition of metabolites from its host. Our study of the cell biology and physiology of a Micrarchaeon and its host adds to our limited knowledge of archaeal symbioses.

[1] Department of Applied Biology, Karlsruhe, Institute of Technology (KIT), Karlsruhe, Germany. [2] Institute for Biological Interfaces, Karlsruhe, Institute of Technology (KIT), Eggenstein-Leopoldshafen, Germany. [3] Sir William Dunn School of Pathology, University of Oxford, Oxford OX1 3RE, United Kingdom. [4] Bioinformatics & Biochemistry, Technische Universität Braunschweig, Braunschweig, Germany. [5] Braunschweig Integrated Centre for Systems Biology (BRICS), Technische Universität Braunschweig, Braunschweig, Germany. [6] Department of Marine Microbiology and Biogeochemistry, NIOZ, Royal Netherlands Institute for Sea Research, Den Burg, The Netherlands. [7] Department of Earth Sciences, Faculty of Geosciences, Utrecht University, Utrecht, The Netherlands. [8] Leibniz Institute DSMZ, Braunschweig, Germany. [9] Helmholtz-Centre for Environmental, Research UFZ, Magdeburg, Germany. [10] Structural Studies Division, MRC Laboratory of Molecular Biology, Francis Crick Avenue, Cambridge CB2 0QH, United Kingdom. [11] Center for Electron Microscopy, University of Regensburg, Regensburg, Germany. [12] Department of Cell- and Molecular Biology, Science for Life Laboratory, Uppsala University, Uppsala, Sweden. [13] Institute of Technical Microbiology, Technical University of Hamburg, Hamburg, Germany. [14] These authors contributed equally: Susanne Krause, Sabrina Gfrerer. ✉email: johannes.gescher@tuhh.de

In 2002, Huber and colleagues described a novel nano-sized archaeon, *Nanoarchaeum equitans*[1]. Later, metagenomic data of environmental samples revealed that the Nanoarchaeota are part of a tentative superphylum of nano-sized archaea referred to as DPANN – an acronym of its first members' lineages, the Diapherotrites, Parvarchaeota, Aenigmarchaeota, Nanoarchaeota, and Nanohaloarchaeota[2]. Most DPANN representatives have reduced genomes and are thought to comprise a diversity of potential archaeal symbionts. Besides the name-giving phyla, the DPANN also include the Woese- and Pacearchaeota[3], Huberarchaeota[4], Micrarchaeota[5], Altiarchaeota[6], Undinarchaeota[7] and Mamarchaeota[8,9] as well as several so far undefined phyla[8,10]. Nano-sized archaea are globally distributed and can comprise non-negligible proportions of microbial communities[11]. Yet only a few representatives have been enriched under laboratory conditions[1,12–17]. Current genomic data suggest that most DPANN archaea have reduced genomes, limited metabolic capabilities and various auxotrophies and might depend on interactions with other community members. The extent of genome reduction varies within the DPANN members. For instance, marine Nanoarchaeota are characterised by highly reduced genomes of about 0.5 Mbp and seem to represent ectoparasites that are strongly dependent on their host[1]. On the other hand, the first members of the Nanohaloarchaeota and Micrarchaeota[5] have larger genome sizes and seem metabolically more flexible[14–16,18]; yet cultivated Nanohaloarchaeota representatives are nevertheless host-dependent[16,17]. While recent work has provided more insights into symbiotic interactions characterising certain representatives of the DPANN[16,19], additional model systems remain to be established. We have recently succeeded in enriching a member of the Micrarchaeota in a community of four different microorganisms[15]. Here, we report the isolation of a stable co-culture of this Micrarchaeon together with its host, a previously unknown member of the *Thermoplasmatales*, as well as the isolation of the latter. This allows us to conduct experiments aiming at understanding the interaction of the two organisms and the response of the *Thermoplasmatales* member to growth in co-culture with the Micrarchaeon.

## Results and discussion

### Isolation of the *Ca.* Micrarchaeum harzensis-*Ca.* Scheffleriplasma hospitalis co-culture and pure culture of *Ca.* Scheffleriplasma hospitalis.

We have previously enriched for a putative DPANN symbiont belonging to the Micrarchaeota, i.e., A_DKE, in a culture, which besides the inferred host *Thermoplasmatales* archaeon B_DKE, also contained a *Cuniculiplasma*-related archaeon referred to as C_DKE and a fungus (with *Acidothrix acidophila* as its closest related organism)[15]. In order to obtain a better understanding of the interactions between the symbiont and its hosts, a co-culture of the putative symbiont and host organisms was generated by eliminating all other organisms. C_DKE represented the minority of the archaea in the enrichment culture and is closely related to an organism reported to have a low pH optimum of 1.0–1.2[20]. Hence, it was possible to eliminate C_DKE by transferring the culture in media with a pH of 2.5, which exceeds its optimal pH range, while still supporting the growth of A_DKE and B_DKE. Secondly, the fungus, which was isolated and via sequence analysis revealed to be most closely related to an isolate from acidic soils in the Czech Republic[21], was successfully eliminated by incubating the enrichment cultures at 37 °C over three consecutive culture transfers. Microscopic analysis, as well as 16S rRNA and ITS region gene analyses confirmed the absence of those contaminants. From the previous microscopy analysis, it was known that A_DKE thrives together with B_DKE in biofilm-like structures[15]. We discovered that it was possible to enhance the biofilm formation of the host by lowering the pH-value of the medium from 2.5 to 2.0, which led to robust biofilm formation of the co-culture (Supplementary Figure 1). At the same time, we were able to isolate the host B_DKE by enriching at pH 2.5 for planktonic organisms. The composition of cultures was verified via PCR with organism-specific primers and CARD-FISH, periodically.

Based on the isolation of the *Thermoplasmatales* host and the reconstruction of complete genome sequences of both organisms, we propose the names "Candidatus Micrarchaeum harzensis" *sp. nov.* (N.L. masc./fem. adj. *harzensis*, pertaining to the German region of the Harz Mountains, where the organism was isolated) and "Candidatus Scheffleriplasma hospitalis" *gen. nov. sp. nov.* (Schef′fler.i.plas′ma. N.L. gen. masc. n. *scheffleri* of Scheffler, named in honour of the geologist Dr. Horst Scheffler and in recognition of his work on mine geology and commitment to our work; Gr. neut. n. *plasma* something shaped or moulded; hos.pi.ta ′lis. L. masc. adj. *hospitalis* relating to a guest, hospitable, referring to its ability to serve as a host for "Candidatus Micrarchaeum harzensis").

Previous attempts to purify mesophilic members of the Micrarchaeota with their respective hosts yielded in relatively diverse enrichments or led to the disappearance of the symbiont after some time of incubation[14,18,19]. Here, we used physiology informed strategies for deselecting against additional community members and show that selection for biofilm formation of the *Thermoplasmatales* host population was the critical factor for obtaining a stable co-culture.

### Characterisation of *Ca.* Micrarchaeum harzensis-*Ca.* Scheffleriplasma hospitalis co-culture.

The pure culture of *Ca.* Scheffleriplasma hospitalis and the *Ca.* Micrarchaeum harzensis-*Ca.* Scheffleriplasma hospitalis co-culture reduced ferric iron (Fe(III)) to ferrous iron (Fe(II)) during growth indicating that *Ca.* Scheffleriplasma hospitalis is a dissimilatory ferric iron reducing organism. During growth of the co-culture *Ca.* Micrarchaeum harzensis was able to reach higher cell densities compared to *Ca.* Scheffleriplasma hospitalis and maintained a higher cell number for several weeks, before dropping under the *Ca.* Scheffleriplasma hospitalis cell number at the end of the growth curve (Fig. 1). Furthermore, *Ca.* Scheffleriplasma hospitalis showed a slower

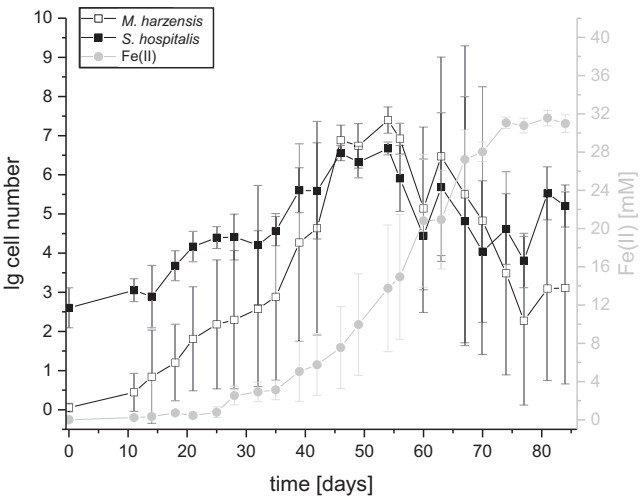

**Fig. 1 Growth curve of a *Ca.* Micrarchaeum harzensis-*Ca.* Scheffleriplasma hospitalis co-culture.** Depicted are the mean cell numbers of *Ca.* Micrarchaeum harzensis (empty squares) and *Ca.* Scheffleriplasma hospitalis (full squares) calculated via qPCR and the ferrous iron concentration (light grey circles) over time. Error bars show standard deviation of triplicates. Source data are provided as a Source Data file.

increase and decrease in cell number compared to the Micrarchaeon.

To examine the impact of the co-cultivation with the Micrarchaeon on *Ca.* Scheffleriplasma hospitalis its growth characteristics were compared between pure and co-culture under otherwise identical conditions (growth curve of *Ca.* Scheffleriplasma hospitalis pure culture in Supplementary Fig. 2). Cells of the *Thermoplasmatales* member showed a similar growth and ferric iron reduction in pure culture in comparison with the co-culture. Also, the doubling time of *Ca.* Scheffleriplasma hospitalis in co-culture ($7.49 \pm 1.45$ days, $n = 3$) was not significantly different (unpaired, two-sample *t*-test, two-sided, significance level 0.05) to the doubling time of the organism cultivated without its interaction partner ($7.19 \pm 3.80$ days, $n = 3$). Hence, growth of the host seems to be little affected by the Micrarchaeon under the tested conditions.

**Genomic potential of *Ca.* Micrarchaeum harzensis and *Ca.* Scheffleriplasma hospitalis**. DNA of the co-culture containing *Ca.* Micrarchaeum harzensis and *Ca.* Scheffleriplasma hospitalis was sequenced using a combination of PacBio and Illumina sequencing. For comparison, Illumina sequencing was also performed on the *Ca.* Scheffleriplasma hospitalis pure culture. The two organisms have circular chromosomes of 1,959,588 base pairs (bp) (*Ca.* Scheffleriplasma hospitalis) and 989,838 bp (*Ca.* Micrarchaeum harzensis), and GC contents of 44.4% and 45.8%, respectively. The genomes of the pure *Ca.* Scheffleriplasma hospitalis isolate and the strain within the co-culture were 100% identical which was important for the later comparative transcriptomic analysis. An analysis of clusters of orthologous groups (COGs) revealed that *Ca.* Micrarchaeum harzensis contains more proteins with unknown function (29%; 300 putative proteins without an arCOG-assignment) relative to the overall number of genes compared to *Ca.* Scheffleriplasma hospitalis (20%; 419 putative proteins without an arCOG-assignment).

The complete genome of *Ca.* Micrarchaeum harzensis confirmed earlier findings[15,18,22], such as an extremely limited set of genes coding for proteins involved in central carbon metabolism. We could only detect one gene encoding a putative enzyme of the pentose phosphate pathway and two genes for putative enzymes of a glycolysis or gluconeogenesis pathway (Supplementary Data 1). However, we did identify a putative set of genes coding for enzymes for the conversion of glucose to glycerate, which together comprise four of the seven reactions of the non-phosphorylative Entner-Doudoroff pathway (Supplementary Data 1). The *Ca.* Micrarchaeum harzensis genome also contains genes encoding enzymes for the conversion of pyruvate to acetyl-CoA, though we could not identify candidate proteins for reactions leading from glycerate to pyruvate. In agreement with our previous study, enzymes for almost all steps of the tricarboxylic acid cycle (TCA) were detected in *Ca.* Micrarchaeum harzensis. Earlier publications proposed that the missing succinyl-CoA synthetase in Micrarchaeota genomes could be replaced by a methylisocitrate lyase, which generates succinate from methylisocitrate[18]. A corresponding gene, encoding methylisocitrate lyase, was detected in the *Ca.* Micrarchaeum harzensis genome, as well (Supplementary Data 1). The thereby potentially completed TCA cycle, may through the production of NADH, fuel an electron transport chain and the generation of a proton gradient for ATP production. In particular, we detected gene clusters encoding a full NADH dehydrogenase and a membrane-bound $A_1/A_0$ ATP synthase complex[23] in the *Ca.* Micrarchaeum harzensis genome. Moreover, we identified genes encoding one subunit of the cytochrome bc1 complex and two subunits of the cytochrome c oxidase (Supplementary Data 1). Although the organism might have the ability to conserve energy and produce reducing equivalents, it will be dependent on building blocks acquired either from the environment (or the culture medium) or from the partner organism *Ca.* Scheffleriplasma hospitalis. For instance, *Ca.* Micrarchaeum harzensis has major gaps in various biosynthesis pathways including for the production of amino acids; the few catalytic steps encoded by *Ca.* Micrarchaeum harzensis comprise the synthesis of aspartate from oxaloacetate, glutamate from α-ketoglutarate and phenylalanine from phenylpyruvate. Phenylpyruvate could be produced from tyrosine, which was taken up from the medium by the co-culture (see below). Other amino acid biosynthesis pathways could not be detected. Furthermore, genes encoding known amino acid transporters seem to be absent[8,10] and DNA, RNA, and lipid biosynthesis pathways are incomplete (see below). Consequently, *Ca.* Micrarchaeum harzensis may acquire certain metabolites or building blocks directly from its partner *Ca.* Scheffleriplasma hospitalis through cell-cell interactions as seen in the *Ignococcus hospitalis-Nanoarchaeum equitans* system[24,25]. In turn, the dependency of *Ca.* Micrarchaeum harzensis on growth in a biofilm (see above) may be due to the need to establish cellular contact with its host. Possible central metabolic pathways were summarised in a cell cartoon in Fig. 2.

**Biofilm composition of pure and co-cultures**. As the isolation experiments and our previous results point towards the importance of extracellular polymeric substances (EPS) for successful cultivation of *Ca.* Micrarchaeum harzensis, we next investigated the composition of the EPS matrix in the co-culture as compared to the pure culture of *Ca.* Scheffleriplasma hospitalis. To this end, the glycoconjugates were analysed with fluorescently labelled lectins, and the signals were correlated to the individual cell type by CARD-FISH analysis (Fig. 3). Lectins are complex proteins, which bind specifically to carbohydrate structures[26]. In this study, 70 different fluorescently labelled lectins, which represent all commercially available lectins (Supplementary Data 2), were used to analyse the EPS in pure and co-cultures.

Among the tested lectins, only those specific to galactose- and mannose-related conjugates bound the extracellular matrix of the co-culture and of *Ca.* Scheffleriplasma hospitalis cells in pure culture (see Fig. 3 and Table 1). Notably, lectins, IAA, HHA, and PTA seemed to discriminate between the extracellular matrix of *Ca.* Scheffleriplasma hospitalis in the presence or absence of *Ca.* Micrarchaeum harzensis, suggesting a potential influence of *Ca.* Micrarchaeum harzensis on the matrix chemistry or composition. Lectin staining of *Ca.* Micrarchaeum harzensis cells was weak and only possible with some of the lectins that bound to *Ca.* Scheffleriplasma hospitalis. This may reflect the inability of *Ca.* Micrarchaeum harzensis to build carbohydrate polymers or the production of less common polymers for which we did not have a lectin. It suggests that the detected signals on *Ca.* Micrarchaeum harzensis are likely due to growth within the biofilm matrix of *Ca.* Scheffleriplasma hospitalis.

Overall, the results indicate that *Ca.* Scheffleriplasma hospitalis displays galactose and mannose on its cell surface and that these carbohydrates are also components of the co-culture EPS matrix. This is corroborated by the presence of transcriptionally expressed genes for metabolic pathways leading to UDP-glucose, UDP-galactose, GDP-mannose, UDP-N-acetylgalactosamine, and UDP-N-acetylglucosamine in the genome of *Ca.* Scheffleriplasma hospitalis (Supplementary Data 3 and Supplementary Fig. 3).

**Membrane lipids of *Ca.* Micrarchaeum harzensis and *Ca.* Scheffleriplasma hospitalis and lipid biosynthetic pathways**. An analysis of the intact polar lipids (IPLs) of co-cultures of *Ca.* Micrarchaeum harzensis and *Ca.* Scheffleriplasma hospitalis revealed the archaeal isoprenoidal glycerol dibiphytanyl glycerol (GDGT) with zero cyclopentane rings (i. e., GDGT-0, also known as

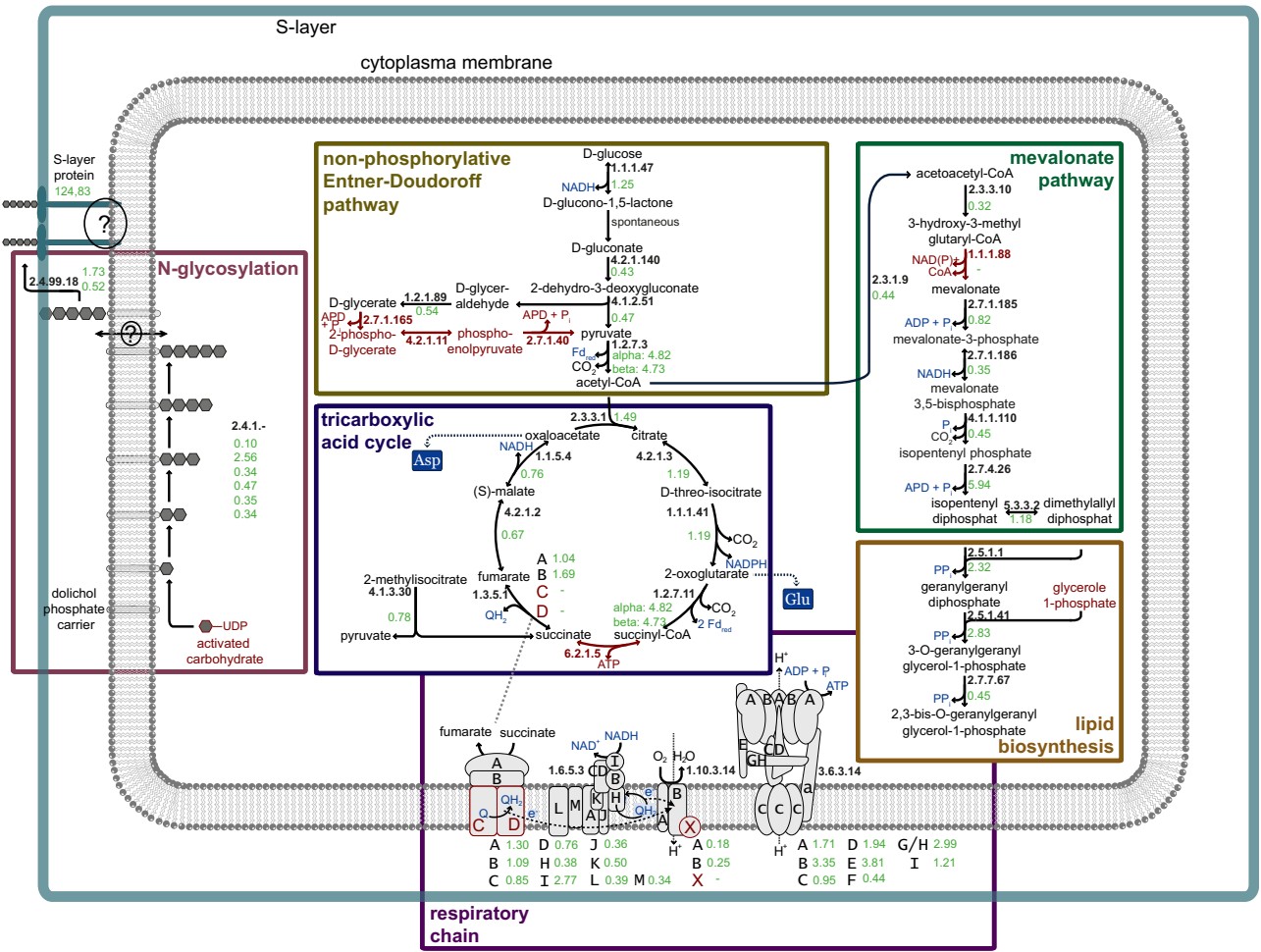

**Fig. 2 Schematic overview of possible central metabolic pathways of *Ca*. Micrarchaeum harzensis.** Given are the enzyme commission (EC) numbers of the involved enzymes in bold and the Transcripts Per Million (TPM) values (mean value of four co-culture samples) of the corresponding *Ca.* Micrarchaeum harzensis genes in green. Missing genes and metabolites that the Micrarchaeon cannot produce itself are highlighted in dark red. Question marks label the unknown anchoring of the S-layer protein and an unknown flippase.

caldarchaeol) as the main lipid, making up to 97% of the total intact polar lipids, together with a minor amount of archaeol (2,3-di-O-phytanyl glycerol diether) (Supplementary Data 4). Comparison of the results to the pure *Ca*. Scheffleriplasma hospitalis-culture revealed no differences in the relative abundance of the archaeal IPLs, suggesting that the Micrarchaeon *Ca*. Micrarchaeum harzensis has an identical membrane lipid composition as its host *Ca*. Scheffleriplasma hospitalis. Archaeal membrane lipids are formed by isoprenoid side chains linked through ether bonds to glycerol-1-phosphate (G1P, synthesised by the G1P-dehydrogenase[27]) either as a bilayer of diethers (archaeols) or a monolayer of tetraethers (i.e., GDGTs). The isoprenoid building blocks are synthesised by one of the four variants of the archaeal mevalonate (MVA) pathway, which differ with regard to the enzymes mediating the last three enzymatic steps (see ref. [28] for a review). Figure 4 shows an overview of the different MVA pathways known. The isoprenoid C20 units are linked to the G1P backbone through ether bonds by the geranylgeranylglyceryl diphosphate (GGGP) synthase and (S)-2,3-di-O-geranylgeranylglyceryl diphosphate (DGGGP) synthase.

The genome data (Supplementary Data 4–6) revealed that *Ca*. Scheffleriplasma hospitalis uses variant-III of the MVA pathways first described in *Thermoplasma acidophilum*[29]. In particular, *Ca*. Scheffleriplasma hospitalis encodes the three key enzymes mevalonate-3-kinase (arCOG02937; M3K), mevalonate-3-

phosphate-5-kinase (COG02074; M3P5K), and mevalonate-3,5-bisphosphate-decarboxylase (arCOG02937; MBD), characterising this pathway, while it lacks genes for a canonical mevalonate kinase (Supplementary Data 5 and 6) similar to other members of the acidophilic *Thermoplasmatales*[30]. Prenyltransferases found in the genome are farnesyl diphosphate synthase and geranylgeranyl diphosphate synthase. Also, a putative G1PDH (i.e., glycerol-1-phosphate dehydrogenase) could be identified. Genes encoding enzymes for the ether bond formation (i.e., GGGP and DGGGP synthase) and saturation of isoprenoids (i.e., geranylgeranyl reductases) are also encoded in the *Ca*. Scheffleriplasma hospitalis genome (see Supplementary Data 6). In contrast, *Ca*. Micrarchaeum harzensis has an incomplete mevalonate and archaeal lipid pathway (Supplementary Data 4 and 5). In particular, while an ancestor of *Ca*. Micrarchaeum harzensis and some other Micrarchaeota are likely to have acquired three key genes of the variant-III mevalonate pathway from *Thermoplasmatales* archaea (i.e., mevalonate-3-kinase (arCOG02937), mevalonate-3-phosphate-5-kinase (COG02074), and mevalonate-3,5-bisphosphate-decarboxylase (arCOG02937)) (Fig. 5), *Ca*. Micrarchaeum harzensis lacks a homologue of the hydroxymethylglutaryl-CoA reductase (Supplementary Data 5). Note, its genome does not provide any evidence for the presence of another variant of the mevalonate pathway (Supplementary Data 5). Furthermore, *Ca*. Micrarchaeum harzensis does not appear

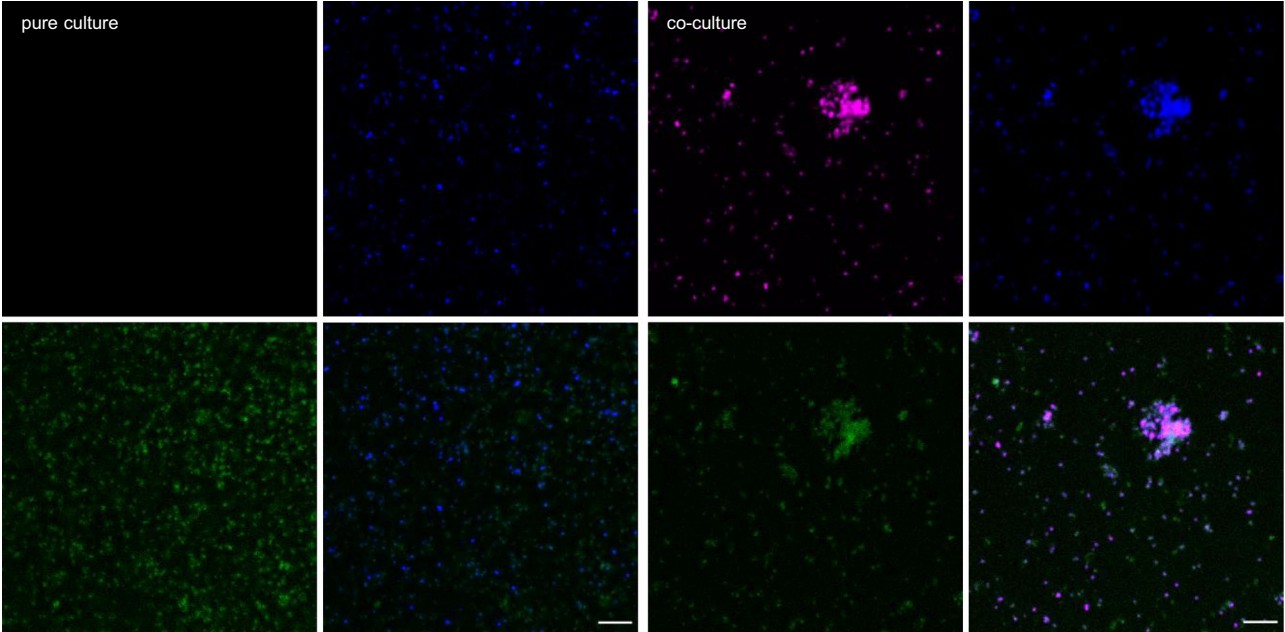

**Fig. 3 Results of lectin staining of co-culture of *Ca*. Micrarchaeum harzensis and *Ca*. Scheffleriplasma hospitalis and *Ca*. Scheffleriplasma hospitalis pure culture.** Shown are example images of the microscopic analysis of the pure culture (left) and co-culture (right). The images show the results for staining with lectin CA (co-culture) and HHA (pure culture). Colour allocation: *Ca*. Scheffleriplasma hospitalis was stained with the general archaea probe Arch915 (blue) which does not stain *Ca*. Micrarchaeum harzensis. *Ca*. Micrarchaeum harzensis was stained using the Micrarchaeota-specific ARMAN980 probe (magenta). Lectin staining is shown in green. Scale bars represent 10 µm. The experiment was repeated at least three times with equivalent results.

**Table 1 Results of lectin staining of co-culture of *Ca*. Micrarchaeum harzensis and *Ca*. Scheffleriplasma hospitalis and *Ca*. Scheffleriplasma hospitalis pure culture. The table shows binding lectins, their abbreviation, carbohydrate-binding specificity and the strength of the signal in pure and co-culture (with + as weak binding, ++ as binding, and − as no binding). The experiment was repeated at least three times with equivalent results.**

| Lectin | Single-sugar binding specifity | *S. hospitalis* pure culture | Co-culture | |
| --- | --- | --- | --- | --- |
| | | | *S. hospitalis* | *M. harzensis* |
| AAL | α-fucose | + | + | − |
| CA | lactose>N-acetylgalactosamine>galactose and related sugars | ++ | ++ | − |
| GNA | mannose | + | ++ | + |
| GS-I | galactose, N-acetylgalactosamine | + | + | − |
| HHA | mannose | ++ | − | + |
| HPA | N-acetylgalactosamine | ++ | + | + |
| IAA | not determined | ++ | ++ | − |
| PTA | galactose, N-acetylgalactosamine | + | − | − |
| RCA | β-galactose, lactose | + | + | + |
| RPA | N-acetylgalactosamine | ++ | ++ | − |
| SSA | α-N-acetylgalactosamine | + | ++ | + |
| TKA | galactose | − | ++ | − |

to encode a G1PDH. One of the encoded geranylgeranyl reductase homologues of *Ca*. Micrarchaeum harzensis, likely involved in lipid biosynthesis, also seems to be acquired by horizontal gene transfer (HGT) from *Thermoplasmatales* (arCOG00570) (Fig. 5). The result is in line with a recent study which revealed that 16.3% of the examined Micrarchaeota genomes contained a horizontally acquired mevalonate-3,5-bisphosphate-decarboxylase[22].

Together with our experimental data, the presence of an incomplete variant-III mevalonate and lipid biosynthesis pathways in *Ca*. Micrarchaeum harzensis, indicates that this organism depends on lipids or precursors thereof from its host, similar to what has been previously described in the DPANN archaeon *N. equitans*[31] and likely other DPANN members such as *Nanohaloarchaeum antarcticus*[16].

**Impact of growth in co-culture on the *Ca*. Scheffleriplasma hospitalis transcriptome**. To further elucidate the effect of the symbiont on its host, we compared gene expression levels of *Ca*. Scheffleriplasma hospitalis with and without co-cultivation with *Ca*. Micrarchaeum harzensis under otherwise identical growth conditions. In particular, we compared three pure cultures with four co-cultures and analysed differentially expressed genes with *p*-values lower than 0.01, a false discovery rate of 0.01 and log2fold changes higher or lower than 2 or −2. This revealed 17 genes that were differentially expressed based on these criteria (Table 2).

Of the 17 differentially expressed genes, 16 genes were downregulated in the host *Ca*. Scheffleriplasma hospitalis in the co-culture compared to the pure culture of *Ca*. Scheffleriplasma

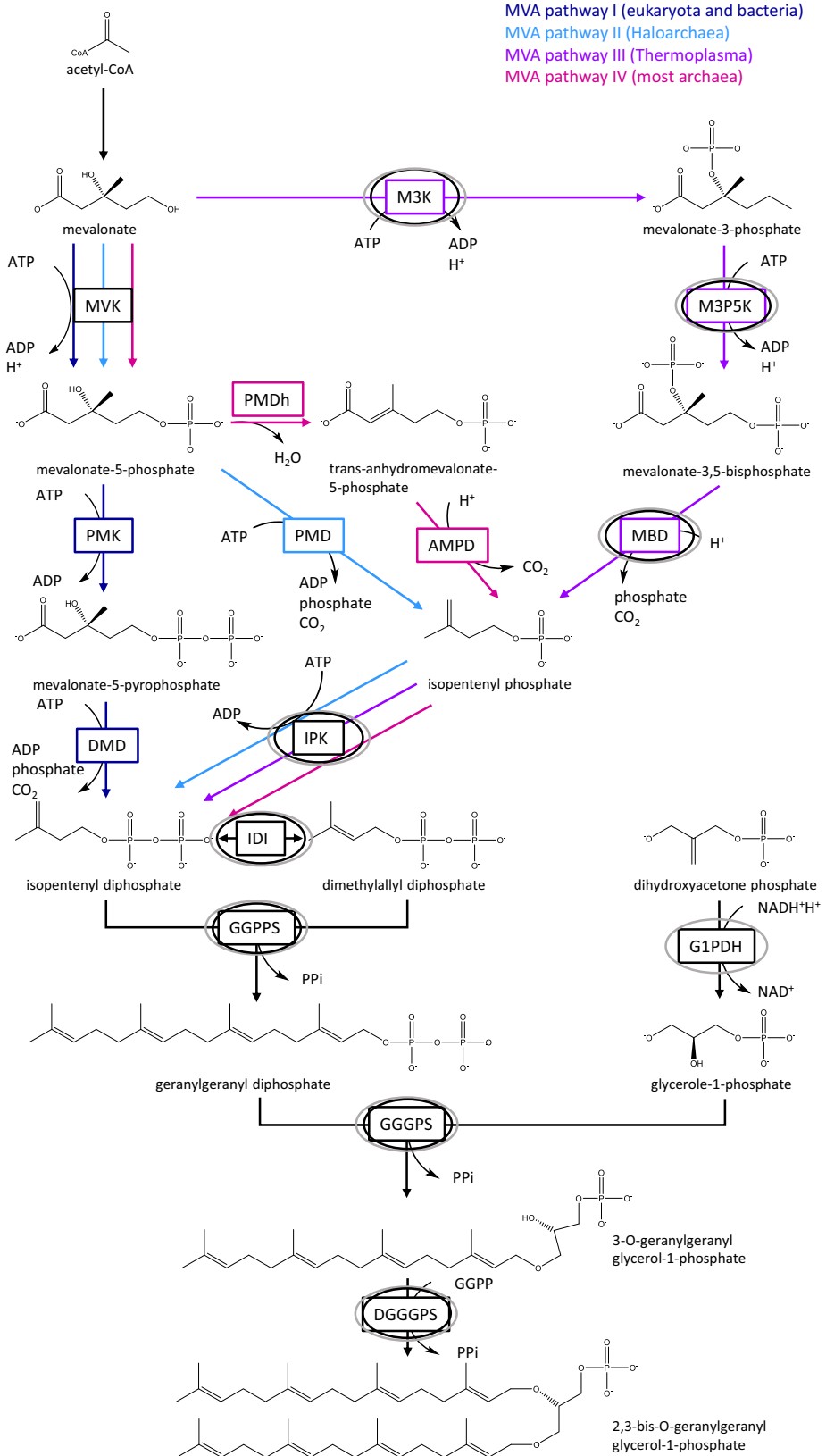

**Fig. 4 Schematic overview of mevalonate pathways and lipid metabolism.** The different pathways are indicated with dark blue (MVA pathway I), light blue (MVA pathway II), violet (MVA pathway III) and magenta arrows (MVA pathway IV), respectively. Names of enzymes are boxed. Enzymes expressed in *Ca.* Micrarchaeum harzensis and *Ca.* Scheffleriplasma hospitalis are indicated with black and grey circles. Abbreviations are AMPD anhydromevalonate phosphate decarboxylase, DGGGPS 2,3-bis-O-geranylgeranyl glycerol-1-phosphate synthase, DMD diphosphomevalonate decarboxylase, GGGPS 3-O-geranylgeranyl glycerol-1-phosphate synthase, GGPP geranylgeranyl diphosphate, GGPPS geranylgeranyl diphosphate synthase, G1PDH glycerol-1-phosphate dehydrogenase, IDI isopentenyl-diphosphate-delta-isomerase, IPK isopentenyl phosphate kinase, M3K mevalonate 3-kinase, M3P5K mevalonate 3-phosphate 5-kinase, MBD mevalonate-3,5-bisphosphate-decarboxylase, MVK mevalonate kinase, PMD phosphomevalonate decarboxylase, PMK phosphomevalonate kinase.

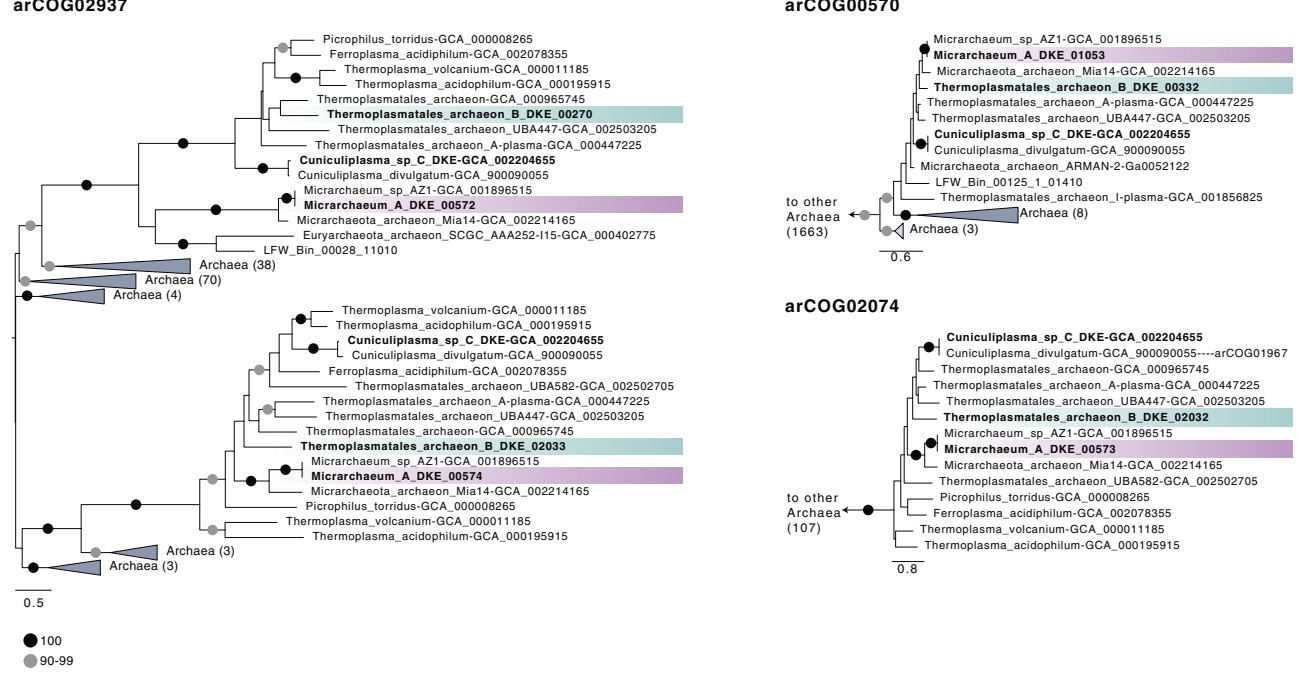

**Fig. 5 Schematic trees of three protein families with indications for gene transfers between acidophilic Micrarchaeota and *Thermoplasmatales*.** *Ca.* Scheffleriplasma hospitalis and *Ca.* Micrarchaeum harzensis genes are highlighted in light blue and violet, respectively. The Maximum likelihood phylogenetic trees shown here included our archaeal backbone dataset (Supplementary Data 7) and were inferred for arCOG02937 (149 sequences with 304 amino acids), arCOG00570 (1877 sequences with 132 amino acids) and COG2074 (arCOG01968 and arCOG01967, 121 sequences with 171 amino acids) with the LG+C10+F+R model with an ultrafast bootstrap approximation run with 1000 replicates. Please note, that the inclusions of bacterial and eukaryotic homologues did not change the interpretation of our findings. Note that arCOG02937 and arCOG02074 represent the key enzymes of the Type-III mevalonate pathway characteristic of *Thermoplasmatales*. Only bootstrap support values above 90 are shown as indicated in the panel.

hospitalis. Five of these genes are part of or are associated with the *Ca.* Scheffleriplasma hospitalis archaellum complex[32]. This suggests a decreased cellular motility of the host in the co-culture, which is in line with the observed tendency of *Ca.* Scheffleriplasma hospitalis to form a biofilm in the presence of *Ca.* Micrarchaeum harzensis. The closest relative of *Ca.* Scheffleriplasma hospitalis, *Cuniculiplasma divulgatum*, does not contain any archaellin-related genes[33], whereas the related 'G-Plasmas' are described to contain the full *arl*-operon (*arlBCDEFGHIJ*)[34]. Furthermore, the gene for the hexuronic acid methyltransferase AglP, a component of the protein glycosylation machinery was downregulated. This may indicate an alteration of the glycosylation pattern of *Ca.* Scheffleriplasma hospitalis, which could influence cell-cell-interaction in the presence of *Ca.* Micrarchaeum harzensis. It may also change the exopolysaccharide (EPS) matrix of *Ca.* Scheffleriplasma hospitalis, which would explain the observed binding differences of some lectins (see above). Three other downregulated genes code for transport proteins that might be involved in the uptake of carbohydrate molecules. While the effect of the decreased expression level of these transporters is unclear, it may be speculated that it could lead to higher availability of certain metabolites in the medium and support growth of *Ca.* Micrarchaeum harzensis. Other downregulated genes encode hypothetical proteins, a putative aminopeptidase, a transposase-associated protein and an iron-sulphur-protein. Thus far, their potential impact on the interaction of *Ca.* Scheffleriplasma hospitalis with *Ca.* Micrarchaeum harzensis remains unclear. Only the gene for a putative membrane protein was upregulated in the co-cultures compared to the pure culture. The encoded protein shows similarities to a protein domain of unknown function (DUF1648), which is part of a receptor protein in *Bacillus subtilis*[35]. Membrane localisation was verified using the

TMHMM algorithm which detected eight transmembrane helices in the protein.

**Metabolomic analysis in the presence and absence of *Ca.* Micrarchaeum harzensis.** Next, we performed a metabolomic analysis to compare metabolites produced in the co-culture to the pure host culture and determine whether the presence of *Ca.* Micrarchaeum harzensis changes the pattern of depleted and produced organic carbon compounds. Growth of both cultures was estimated based on the change in ferrous iron concentration over time. *Ca.* Scheffleriplasma hospitalis pure culture showed in these experiments a shorter lag-phase compared to the co-culture (Supplementary Fig. 4). Therefore, we compared samples with equal ferrous iron concentration (3, 4, and 5 weeks of growth for the pure culture; 5, 6, and 7 weeks of growth for the co-culture) as they indicate similar growth phases. Figure 6 shows a heatmap representing all significantly different metabolites ($p < 0.05$ false discovery rate, FDR corrected) present in the samples at the various time points. While the majority of the detected compounds changed simultaneously in the pure- and the co-culture throughout the growth phases, some metabolites showed different patterns.

Tyrosine levels were found to decrease faster in the co-culture, whereas 4-hydroxyphenyl acetic acid, an intermediate of the tyrosine degradation pathway, increased. Hence, tyrosine degradation seems to be accelerated in the co-culture. Both *Ca.* Micrarchaeum harzensis and *Ca.* Scheffleriplasma hospitalis encode enzymes catalysing the conversion of tyrosine to 4-hydroxyphenylpyuvate (Supplementary Data 1) and it seems possible that the presence of the Micrarchaeon leads to faster tyrosine depletion. Of note, the specific genes for the carboxylase

**Table 2 Differentially expressed genes of *Ca*. Scheffleriplasma hospitalis in pure culture in comparison to *Ca*. Scheffleriplasma hospitalis in co-culture with *Ca*. Micrarchaeum harzensis. The table indicates the gene ID, the description of the expressed protein, the log2-fold change and the *p*-value (<0.01; two-tailed Wald test, Bonferroni corrected).**

| ID | Description | log2-foldchange | *p*-value |
|---|---|---|---|
| Thermo_01860 | Hypothetical protein | −5,41 | 0 |
| Thermo_01798 | Hypothetical protein | −5,23 | 0 |
| Thermo_00025 | Hypothetical protein | −4,87 | 0 |
| Thermo_00444 | 41 kDa archaellin | −4,44 | 0 |
| Thermo_01859 | 41 kDa archaellin | −4,37 | 0 |
| Thermo_01877 | Hypothetical protein | −4,05 | 0 |
| Thermo_00445 | Putative archaellin-related protein C | −3,98 | 0 |
| Thermo_00680 | Putative aminopeptidase 1 | −3,96 | 0 |
| Thermo_00502 | Hexuronic acid methyltransferase AglP | −3,83 | 0 |
| Thermo_00446 | Putative archaellin-related protein D/E | −3,58 | 0 |
| Thermo_01799 | Hypothetical protein | −3,23 | 0 |
| Thermo_01144 | Pink FeS protein | −3,03 | 0 |
| Thermo_00581 | Trehalose/maltose import ATP-binding protein MalK | −2,22 | 0 |
| Thermo_01800 | Hypothetical protein | −2,22 | 4,1592E−08 |
| Thermo_01181 | Trehalose/maltose import ATP-binding protein MalK | −2,17 | 0 |
| Thermo_00582 | ABC-type transport system involved in multi-copper enzyme maturation, permease component | −2,12 | 0 |
| Thermo_01732 | Putative membrane protein | 2,12 | 0 |

and dehydrogenase reaction from 4-hydroxyphenylpyruvate to 4-hydroxyphenyl acetic acid are currently unknown. While 2-phenylglycine, a degradation product of phenylalanine that enters the ketoadipate pathway, decreased, muconic acid, an intermediate of that pathway, increased in the co-culture. Still, we could neither identify genes involved in the degradation of 2-phenylglycine nor for a complete ß-ketoadipate pathway in *Ca*. Micrarchaeum harzensis or *Ca*. Scheffleriplasma hospitalis (Supplementary Data 1). Moreover, the ß-ketoadipate pathway operates under oxic conditions and the organisms were cultivated in the absence of oxygen. Hence, so far, we cannot explain the consumption and production of 2-phenylglycine and muconic acid, respectively. The analyses also revealed increased levels of gluconic acid in the co-culture, which may be a product of sugar-degradation, for instance from the biofilm EPS matrix[36]. Genomic information indicates that *Ca*. Scheffleriplasma hospitalis is able to degrade glucose into glucono-1,5-lactone (KO: K18124), which can spontaneously be converted into gluconic acid (Supplementary Data 1). Both organisms possess the enzymes to convert gluconic acid to glycerate, as discussed above. Overall, our results reveal that the pattern of metabolites do not seem to deviate between isolated *Ca*. Scheffleriplasma hospitalis and co-culture and that the kinetics of consumption show minor differences towards faster consumption of some compounds in the co-culture. Hence, either both organisms employ similar pathways and compounds or, perhaps more likely, *Ca*. Micrarchaeum harzensis predominantly uses metabolites provided by *Ca*. Scheffleriplasma hospitalis. The latter assumption is in agreement with the sparsity of transporters encoded in the *Ca*. Micrarchaeum harzensis genome, which may indicate that it relies on direct cell-cell interaction for nutrient and metabolite exchange.

**Evidence for direct cell-cell interactions between *Ca*. Micrarchaeum harzensis and *Ca*. Scheffleriplasma hospitalis.** Due to the pleomorphic morphology and great variability in cell size of members of the *Thermoplasmatales* including *Ca*. Scheffleriplasma hospitalis, it was previously challenging to clearly distinguish symbiont and host cells on electron micrographs. Recently, Gfrerer et al. revealed that *Ca*. Micrarchaeum harzensis cells are characterised by the presence of an S-layer that can be observed on

electron micrographs of freeze-etched, Platinum-Carbon shadowed samples[37]. Here, using electron microscopy, we could show the attachment of several *Ca*. Micrarchaeum harzensis to *Ca*. Scheffleriplasma hospitalis cells, suggesting direct cell-cell interactions between these organisms (Fig. 7a), as was previously shown for *N. equitans* and *I. hospitalis*[24,25] and in environmental studies containing Micrarchaeota[38]. However, we also observed a large number of *Ca*. Micrarchaeum harzensis cells that were not in contact with their potential host organism (Fig. 7b), which is in agreement with observations from microscopic images of CARD-FISH-stained cultures. While we cannot exclude that this is (to some degree) a result of sample preparation, it is possible that growth in the biofilm enables a more dynamic interaction between *Ca*. Micrarchaeum harzensis and *Ca*. Scheffleriplasma hospitalis than observed for *N. equitans* and *I. hospitalis*[24,25], as the risk of detaching from the host is mitigated by growth within the biofilm matrix. Moreover, *Ca*. Micrarchaeum harzensis has a larger genome and in turn greater metabolic flexibility than *N. equitans*[39] and may in turn be less dependent on permanent attachment to host cells. Finally, we detected several unattached *Ca*. Micrarchaeum harzensis cells in the process of cell division (Fig. 7c). This could either be an artefact of sample preparation or suggest that *Ca*. Micrarchaeum harzensis can store a sufficient amount of building blocks to divide without being in direct cell-cell contact with *Ca*. Scheffleriplasma hospitalis.

**Cryo-electron tomography (cryo-ET) shows attachment sites of *Ca*. Micrarchaeum harzensis and *Ca*. Scheffleriplasma hospitalis.** To further investigate the physical interaction and characteristics of the two organisms, we used cryo-ET investigations of specimens preserved in vitreous ice in a near-native environment. Cryo-ET images of *Ca*. Micrarchaeum harzensis showed a cell membrane surrounded by a prototypical archaeal S-layer[40] with clustered ribosomes abundant in the cytoplasm (Fig. 8a). We interpret the ribosome-free space as 'nucleoid', most likely containing the DNA. Cryo-ET further showed that *Ca*. Scheffleriplasma hospitalis cells are surrounded by a single cell membrane (Fig. 8b). Its cytoplasm is not homogeneous. At the sites of interactions of the two species, we detected filamentous structures between host and Micrarchaeon cells, which appear to contact the S-layer of *Ca*. Micrarchaeum harzensis cells but do not penetrate it (Fig. 8c). We also detected surface remodelling of

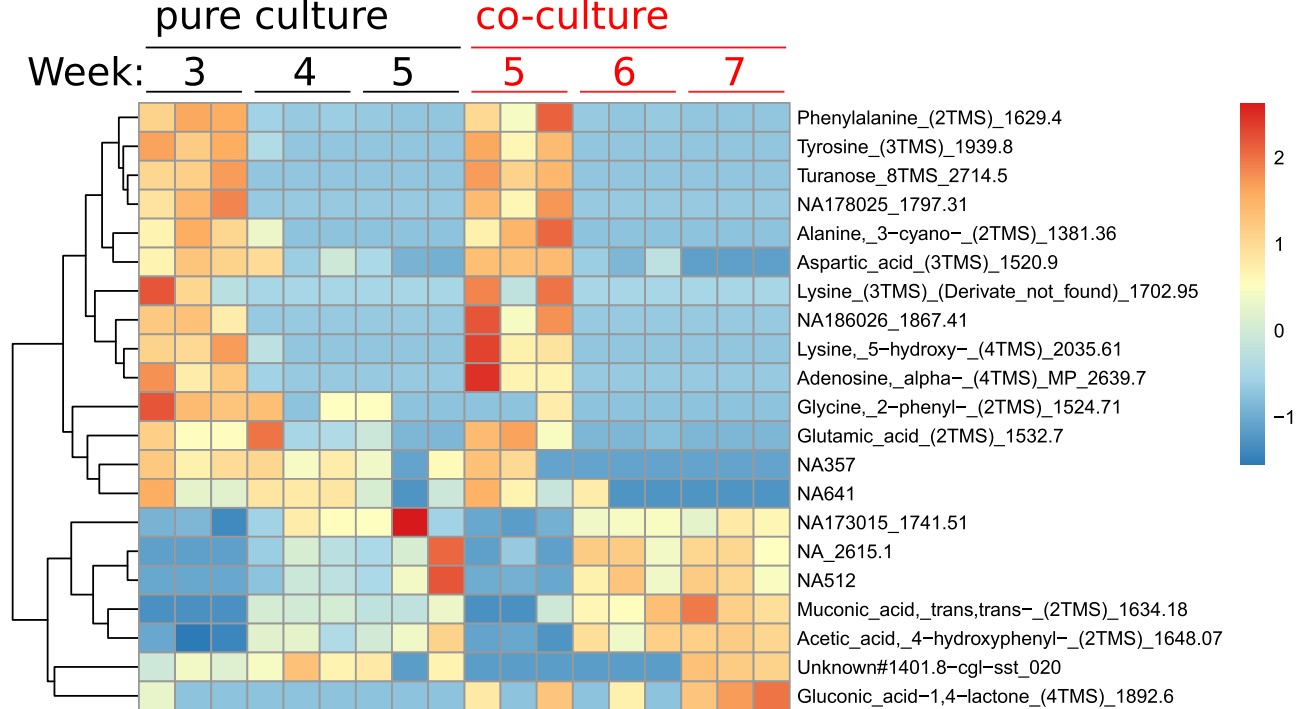

**Fig. 6 Heatmap showing significantly different metabolite levels (*p* < 0.05 false discovery rate, FDR corrected) between corresponding growth phases of pure culture (*Ca*. Scheffleriplasma hospitalis) and co-culture (*Ca*. Micrarchaeum harzensis-*Ca*. Scheffleriplasma hospitalis).** Normalisation was done using a z-score, and significance was calculated by a two-tailed *t*-test. Source data are provided as a Source Data file.

Micrarchaeota and *Ca*. Scheffleriplasma hospitalis cells, which were in direct contact with each other, specifically at the attachment site (Fig. 8d). This suggests that interaction of both organisms might be established by small filamentous structures leading to surface remodelling of the interacting cells. Direct contacts between the cytoplasm of both interaction partners as described for *N. equitans* and *I. hospitalis*[24,25] were not observed in the investigated samples so far. We addressed the question whether *Ca*. Micrarchaeum harzensis is able to divide independently and performed quantitative analysis by ultrastructural characterisation of associated and un-associated dividing Micrarchaeota cells. An analysis of a high-throughput cryo-EM dataset (2,000 images) revealed 77.8% of the dividing cells are associated with a host cell (Fig. 8e) supporting the hypothesis that *Ca*. Micrarchaeum harzensis predominantly relies on host interactions for cell division.

**Integration of results reveals a specific regulatory response towards co-culturing providing future research question.** The detailed characterisation of our co-culture in comparison with pure host cultures indicate a specific regulatory response of *Ca*. Scheffleriplasma hospitalis as a consequence of growth with *Ca*. Micrarchaeum harzensis. The combination of genomic analyses with comparative metabolomics, lipidomics and determination of EPS composition revealed that the growth of *Ca*. Micrarchaeum harzensis is dependent on interaction with its host *Ca*. Scheffleriplasma hospitalis, whose growth seemed little impaired by the symbiont. Cryo-ET micrographs and comparative transcriptomics indicate that *Ca*. Scheffleriplasma hospitalis might initiate the interaction by biofilm formation, change of EPS composition and development of a filamentous structure, which was observed in contact with Micrarchaeota cells. A physical interaction involving surface remodelling is established, which is likely crucial for the uptake of various metabolites and building blocks for, among others, membrane formation. Due to the limited metabolic capabilities of *Ca*.

Micrarchaeum harzensis and its dependency on *Ca*. Scheffleriplasma hospitalis for growth, it is very likely that the DPANN member initiates the interaction. The close cell-cell interactions between acidophilic Micrarchaeota and *Thermoplasmatales* may also provide a route for horizontal gene transfer among these DPANN symbionts and their hosts.

It will be interesting to compare cell-cell interactions underlying this system to those characterising the various other members of the extremely diverse DPANN archaea and establish unique and common characteristics[12–14,16,17,41]. To this end, similar observations were published while this manuscript was under consideration[41]. A thermoacidophilic co-culture between a Micrarchaeon and a new isolate belonging to the genus *Metallosphaera* also revealed direct interactions between the Micrarchaeon and its host. Nevertheless, the results also reveal the physiological variability of Micrarchaeota members, as the co-culture enriched by Sakai et al. was not derived from a biofilm but acidic spring water and grew aerobically. Further studies will reveal whether these differences can be traced back to genomic or transcriptomic variations. On a broader scale, insights into symbiont-host interactions involving DPANN representatives will be important to improve our understanding of their role in ecological networks, considering their prevalence in most environments on Earth[2,8,10,42].

## Methods

**Culturing conditions.** The pure culture of *Ca*. Scheffleriplasma hospitalis (JCM 39074) and the co-culture of *Ca*. Scheffleriplasma hospitalis and *Ca*. Micrarchaeum harzensis were cultivated under anoxic conditions in a modified *Picrophilus* medium[15] at a pH of 2.0 or 2.5 and at 22 °C. The pH of *Picrophilus* salt solution was adjusted with $H_2SO_4$ and oxygen was eliminated from the solution via boiling and subsequent exchanging of the headspace with $N_2$. After autoclaving the medium was complemented with anoxic, sterile filtered solutions of yeast extract (final concentration 0.1%), casein hydrolysate (final concentration 0.1%) and ferric sulphate (final concentration 20 mM). Cultivation took place in anoxic flasks with a headspace containing 5% $H_2$ and 95% $N_2$. Transfers of the cultures were conducted with 20% of the pre-culture. The growth phase was assessed by following the

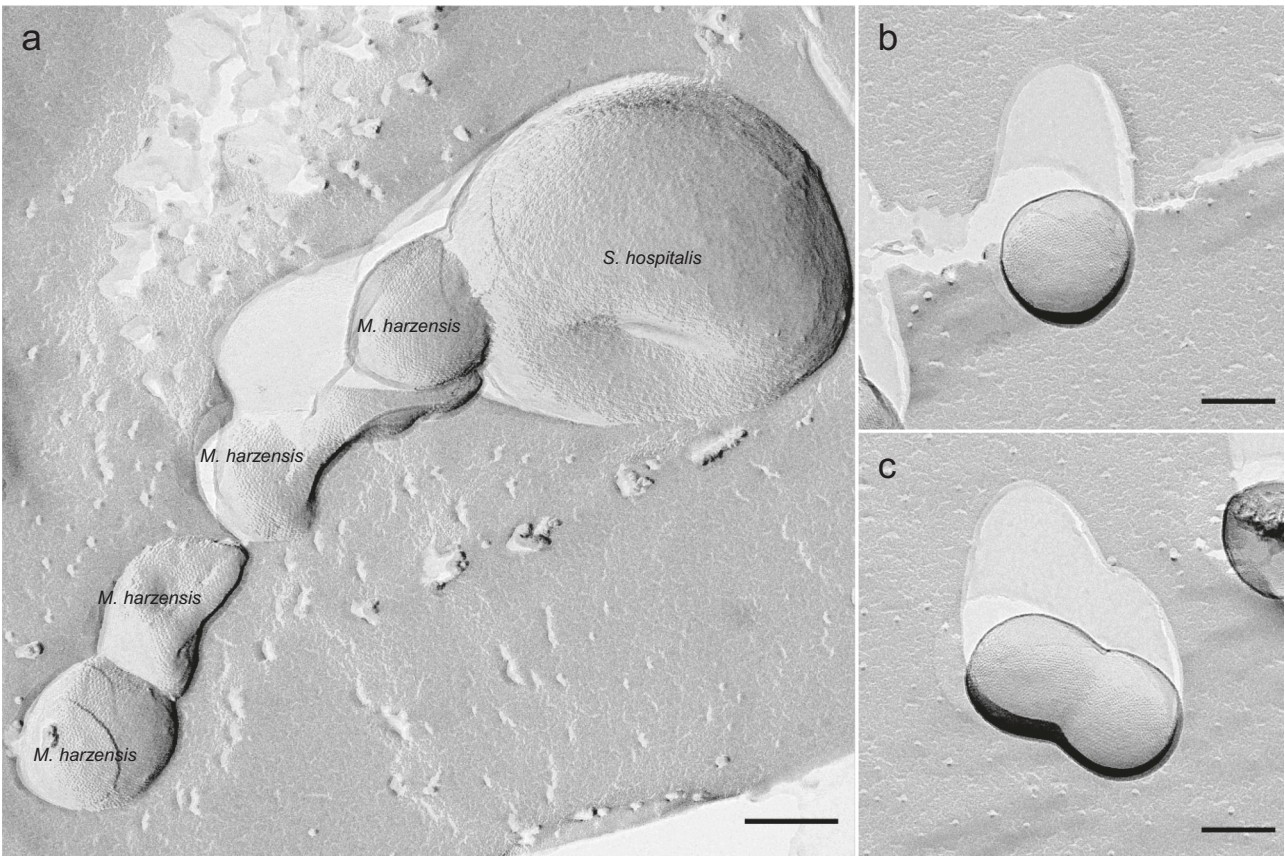

**Fig. 7 Electron micrographs of freeze-etched, Platinum-Carbon shadowed co-culture cells, containing *Ca*. Scheffleriplasma hospitalis and *Ca*. Micrarchaeum harzensis.** *Ca*. Micrarchaeum harzensis cells, displaying an S-layer on their surface, were observed (**a**) in physical contact with *Ca*. Scheffleriplasma hospitalis cells, as well as (**b**) free living and (**c**) undergoing cell division. Scale bars equal 200 nm. The freeze-etching experiment was repeated once, each time micrographs showing more than 100 cells were analysed.

reduction of ferric iron via the ferrozine assay[43]. Furthermore, the cultures were monitored regarding their activity via CARD-FISH (see below) and regarding their composition via PCR with organism-specific primers (Supplementary Data 9). The cultures typically reached the exponential growth phase after incubation for 2–8 weeks.

**CARD-FISH and lectin staining**. Samples were fixed for 1 h in 4% formaldehyde, washed twice in phosphate-buffered saline (PBS), and stored at −20 °C in 50:50 PBS/ethanol for CARD-FISH analysis[15,44] or at 4 °C in 100% PBS for lectin staining. The fixed cells were immobilised on a PTFE-coated slide (Thermo Fisher Scientific Inc., Schwerte, Germany). Inherent peroxidases were inactivated via a 10 min incubation in 0.1 M HCl, followed by a quick wash step with ddH$_2$O. Hybridisation of specific probes was carried out via incubating the dried cells with 0.3 ng/µL HRP-labelled probe in appropriate hybridisation buffer[44] for 1.5 h at 46 °C followed by three wash steps (quick dip in appropriate washing buffer, 15 min at 48 °C in washing buffer, 15 min in saline sodium citrate (SSC) buffer). Amplification of signals involved a 15 min incubation of cells at 37 °C with 0.5 µL fluorophore-coupled tyramide (Thermo Fisher Scientific Inc., Schwerte, Germany) in 500 mL amplification buffer (4 mL 10 x PBS, 16 mL 5 M NaCl, 4 g dextran sulphate, 0.4 mL 10% blocking reagent (Sigma-Aldrich, Steinheim, Germany), 19.6 mL ddH$_2$O) mixed with 5 µL H$_2$O$_2$. For subsequent wash steps incubation of the slide for 15 min in SSC was repeated once followed by a quick dip in ddH$_2$O. For double hybridisation the previous steps were repeated starting with the inactivation of peroxidases. Thereafter, the dried slides were treated with DAPI solution (1 mg/mL) for 5 min and subsequently incubated at ddH$_2$O for 2×1 min. At last, cells were mounted in embedding buffer (10 mL Citifluor (Citifluor Limited, London, UK), 2 mL Vectashield (Vector Laboratories, Burlingame, CA, USA), 1 mL phosphate-buffered saline (PBS)) and stored at −20 °C until further analysis. Labelling was conducted using HRP-labelled 16S rDNA probes ARCH915 (Archaea domain, GTG CTC CCC CGC CAA TTC CT, 20% formamide, purchased at biomers.net GmbH, Ulm, Germany;[45]), TH1187 (*Thermoplasmatales*, GTA CTG ACC TGC CGT CGA C, 20% formamide, purchased at biomers.net GmbH, Ulm, Germany;[46]) and ARM980 (ARMAN, GCC GTC GCT TCT GGT AAT, 30% formamide, purchased at biomers.net GmbH, Ulm, Germany;[47]). For standard CARD-FISH staining, Alexa546 and Alexa488 fluorophores were used,

and counterstaining was conducted with DAPI. CARD-FISH staining for lectin analysis was conducted using Alexa546 and Alexa647 fluorophores; see below for more details. Slides were visualised on a Leica DM 5500B microscope (objective lens 100×: HCX PL FLUOTAR, 1.4, oil immersion and objective lens 64×: HCX PL APO; eyepiece 10×: HC PLAN s (25) M), and images were taken with a Leica DFC 360 FX CCD camera and the corresponding Leica LAS AF 6000 software.Lectin staining was conducted according to a protocol of ref. [48]. The positive lectin results were compared with CARD-FISH staining of the same slides, using a protocol of ref. [49]. CARD-FISH staining was performed as described above with the following modifications: 1) all ethanol washing steps were omitted, as these would negatively affect the lectin staining[49]; 2) cells were not counterstained with DAPI. After fixation and staining via CARD-FISH, the samples were dried at 37 °C, and 100 µL lectin solution (0.1 mg/mL) was added and incubated for 30 min at room temperature in the dark. After washing with PBS solution, the slides were dried at 37 °C and mounted in the same embedding buffer as used for CARD-FISH. Please refer to Supplementary Data 2 for an overview of lectins used. Imaging was conducted using a Zeiss Axiovert 200 M fluorescence microscope equipped with the software Axio Vision 4.7.

**DNA/RNA isolation and quantitative PCR analysis**. Isolation of DNA for quantitative (qPCR) analysis was conducted with the Invisorb Spin Forensic Kit following the manufacturer's instructions (Invitek, Berlin, Germany). DNA for Illumina and PacBio sequencing was isolated as described by ref. [50]. In total 50 mL of three co-cultures were spun down at 15,000 × g for 10 min. 3 mL of Tris-EDTA (50 mM Tris-HCl pH 7.5, 50 mM EDTA pH 8.0) was used for resuspending of the cells. After addition of 0.6 mL STEP buffer (0.5% SDS, 50 mM Tris-HCl pH 7.5, 400 mM EDTA, 1 mg/mL proteinase K, 0.5% sarkosyl) the lysate was incubated at 50 °C overnight. Subsequent phenol-chloroform extraction involved the addition of 3.6 mL phenol-chloroform isoamyl and hand-shaking for 5 min followed by a centrifugation step at 6000 × g for 10 min. The water fraction was transferred to another tube and the extraction procedure was repeated for a total of three times. Precipitation of DNA was conducted by adding an equal volume of isopropanol and 0.1× the volume of sodium acetate (3 M) to the solution and incubation at −20 °C overnight. Precipitated DNA was centrifuged at 15,000 × g for 30 min, washed once with 30% ethanol (30 min 15,000 × g) and dried at room temperature,

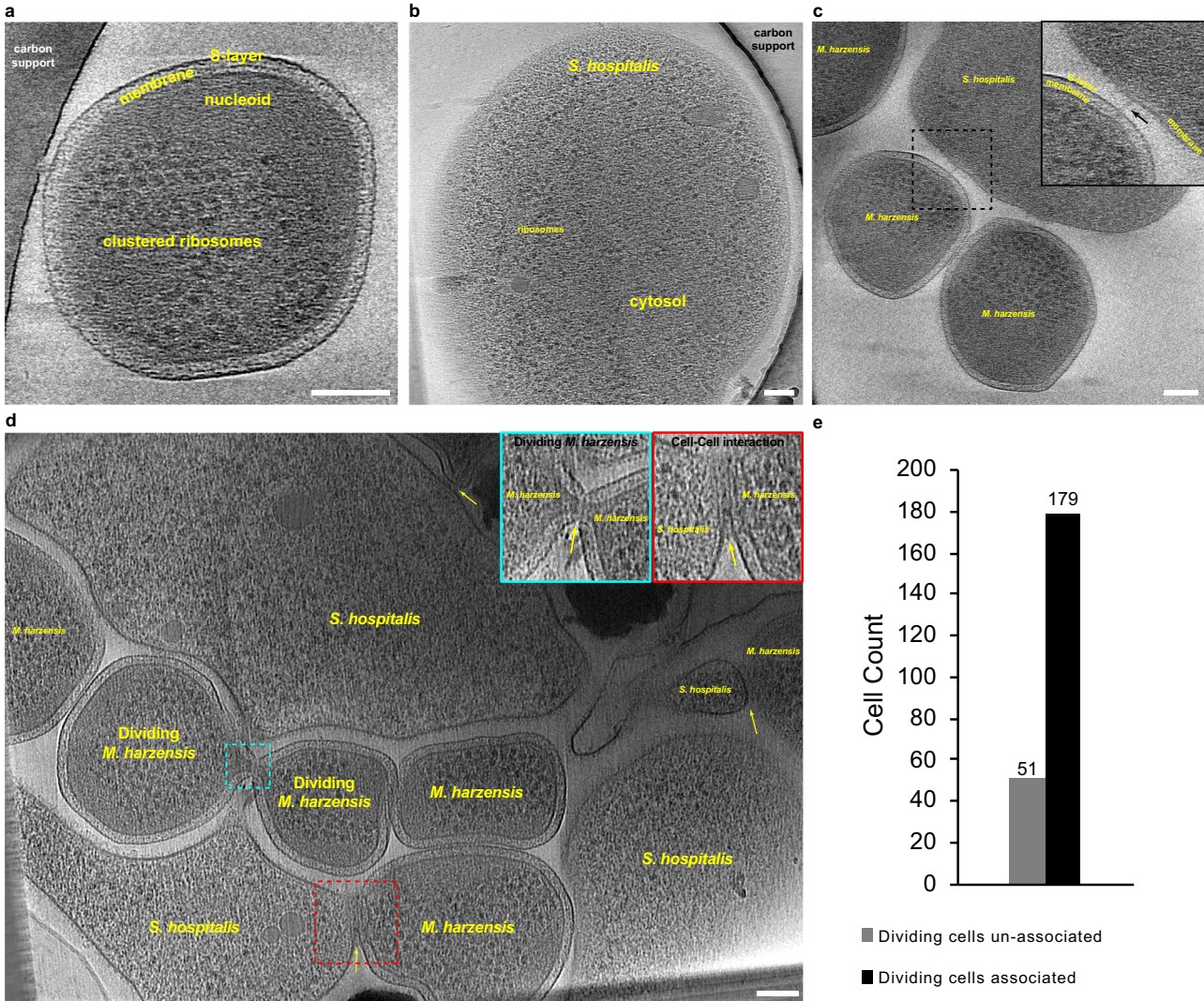

**Fig. 8 Cryo-ET study of a *Ca.* Micrarchaeum harzensis-*Ca.* Scheffleriplasma hospitalis co-culture.** Shown are (**a**) a tomographic slice through a *Ca.* Micrarchaeum harzensis cell, (**b**) a tomographic slice through a *Ca.* Scheffleriplasma hospitalis cell, (**c**) attachment sites and (**d**) cell surface remodelling of both organisms, as well as (**e**) results of a quantification of associated and un-associated dividing *Ca.* Micrarchaeum harzensis cells. Arrows highlight a filamentous structure (panel **c**) and attachment sites undergoing surface remodelling (panel **d**). Scale bars equal 100 nm. Additional Tomograms displaying *Ca.* Micrarchaeum harzensis and *Ca.* Scheffleriplasma hospitalis cells, as well as the corresponding movie to Fig. 8c are part of the supplements (Supplementary Movies 1–3). The experiment was performed two times with equivalent results.

before subsequent resuspension in TE buffer at 4 °C overnight. RNA isolation and library preparation was conducted by IMGM laboratories GmbH using the RNeasy Micro Kit (Qiagen, Hilden, Germany) and TruSeq® Stranded total RNA LT kit according to the manufacturer's instructions (Illumina, Berlin, Germany). All sequencing analyses were conducted with samples of exponentially growing pure host cultures as well as symbiont-host co-cultures. The cell number of the Micrarchaeon *Ca.* Micrarchaeum harzensis and the *Thermoplasmatales* archaeon *Ca.* Scheffleriplasma hospitalis was calculated via qPCR using standard curves of a modified *E. coli* with the primer target sites integrated to its genome[15]. The primer sequences are listed in Supplementary Data 9.

**Metagenome analysis.** A SMRTbell™ template library was prepared according to the instructions from PacificBiosciences, Menlo Park, CA, USA, following the Procedure & Checklist – Greater Than 10 kb Template Preparation. Briefly, for preparation of 15 kB libraries, DNA was end-repaired and ligated overnight to hairpin adaptors applying components from the DNA/Polymerase Binding Kit P6 from Pacific Biosciences, Menlo Park, CA, USA. Reactions were carried out according to the manufacturer's instructions. BluePippin™ Size-Selection to greater than 4 kb was performed according to the manufacturer's instructions (Sage Science, Beverly, MA, USA). Conditions for annealing of sequencing primers and binding of polymerase to purified SMRTbell™ template were assessed with the Calculator in RS Remote, PacificBiosciences, Menlo Park, CA, USA. One SMRT cell was sequenced on the PacBio RSII (PacificBiosciences, Menlo Park, CA, USA),

taking one 240-min movie. Libraries for sequencing on the Illumina platform were prepared to apply the Nextera XT DNA Library Preparation Kit (Illumina, San Diego, USA) with modifications according to ref. [51] and sequenced on Illumina NextSeq™ 500 (Illumina, San Diego, USA).

Genome assembly was performed by applying the RS_HGAP_Assembly.3 protocol included in the SMRT Portal version 2.3.0. The assembly revealed two major contigs. Potentially misassembled artificial contigs with low coverage and included in other replicons were removed from the assembly. Redundancies at the ends of the two major contigs allowed them to be circularised. Replicons were adjusted to *smc* (chromosome partition protein Smc) as the first gene. Error-correction was performed by mapping of the Illumina short reads onto finished genomes using the Burrows-Wheeler Aligner bwa 0.6.2 in paired-end mode using default settings[52] with subsequent variant and consensus calling using VarScan 2.3.6 (Parameters: mpileup2cns --min-coverage 10 --min-reads2 6 --min-avg-qual 20 --min-var-freq 0.8 --min-freq-for-hom 0.75 --p-value 0.01 --strand-filter 1 --variants 1 --output-vcf 1)[53]. Automated genome annotation was carried out using Prokka 1.8[54].

**Genome annotations.** For further functional annotation, the protein files from the two complete genomes as well as 22 Micrarchaeota and 11 *Thermoplasmatales* reference genomes (Supplementary Data 8) were compared against several databases, including the Archaeal Clusters of Orthologous Genes (arCOGs[55]; version from 2018), the KO profiles from the KEGG Automatic Annotation Server

(KAAS[56]; downloaded April 2019), the Pfam database ([57] Release 31.0), the TIGRFAM database[58] (Release 15.0), the Carbohydrate-Active enZymes (CAZy) database ([59] downloaded from dbCAN2 in September 2019), the Transporter Classification Database (TCDB[60]; downloaded in November 2018), the hydrogenase database (HydDB[61]; downloaded in November 2018) and NCBI_nr (downloaded in November 2018). Hmmsearch v3.1b298 was used to read HMM profiles of the ArCOG, PFAM, TIGRFAM and CAZyme databases and search against a protein database (settings: hmmsearch <hmmfile><seqdb> –E 1e–4[62]). The best hit for each protein was selected based on the highest e-value and bitscore by using a custom script (hmmsearchTable, available at https://zenodo.org/record/3839790[7]). BlastP was used with TCBD, HydDB and NCBI_nr as input databases and the protein sequences as query (settings: -evalue 1e-20 -outfmt 6). Additionally, all proteins were scanned for protein domains using InterProScan (v5.29-68.0; settings: --iprlookup –goterms[63]). For InterProScan we report multiple hits corresponding to the individual domains of a protein using a custom script (parse_IPRdomains_vs2_GO_2.py) (Supplementary Data 5 and 6). All custom scripts are available at https://zenodo.org/record/3839790[7].

**Phylogenetic analyses.** We performed phylogenetic analyses of membrane lipid biosynthetic proteins, whenever homologues of relevant arCOGs were present in both *Ca*. Micrarchaeum harzensis and *Ca*. Scheffleriplasma hospitalis to assess the extent of horizontal gene transfer (HGT) affecting these proteins (Supplementary Data 7). In particular, we extracted homologues of corresponding arCOGs for all of these proteins from *Ca*. Micrarchaeum harzensis, *Ca*. Scheffleriplasma hospitalis, a reference set of 566 archaeal genomes (archaea-only analysis) as well as from an additional of 3020 bacterial and 100 eukaryotic genomes (universal analysis) (Supplementary Data 8). The reference genomes were annotated as described above. For the archaea-only analysis, the individual homologues for each protein family were aligned using MAFFT L-INS-i v7.407 (settings: --reorder[64]), trimmed with BMGE v1.12 (settings: -t AA -m BLOSUM30 -h 0.55[65]). Subsequently, phylogenetic trees were inferred using IQ-TREE (v1.6.10, settings: -m LG+C10+F +R -wbtl -bb 1000 -bnni[66]). For the universal analysis, MAFFT L-INS-i v7.407 and MAFFT v7.407 were used to align protein families with Less/equal (≤) or more (>) than 1000 homologues, respectively. BMGE v1.12 was used for trimming all alignments (settings: -t AA -m BLOSUM30 -h 0.55[65]) and phylogenetic trees were inferred using IQ-TREE (v1.6.10, settings: -m LG+C10+F+R -wbtl -bb 1000 -bnni[66]). Due to the large number of sequences affiliating with arCOG00570 when including bacterial and eukaryotic homologues (i.e., 13381), sequences for this protein family were aligned using MAFFT v7.407, trimmed with TrimAL (v1.2rev59, settings: -gappyout[67]). Sequences with ≥90% gaps were removed using a custom script (faa_drop.py) and used for a phylogenetic analysis with FastTree (v2.1.10, settings: -lg -gamma).

**Transcriptomic analysis.** Sequencing of RNA samples of three replicates of the pure culture and four replicates of the co-culture was performed on an Illumina NextSeq® 500 NGS system using 2 ×75 bp paired-end read chemistry. All samples were taken from cultures in exponential growth phase. Transcriptomic analyses were performed using Kallist v0.45.0[68] and compared to the reference completed genomes (see above). Differential expression was assessed by using the Create Expression Browser 1.1 implementation of CLC Genomics Workbench 20.0.1 (QIAGEN, Aarhus, Denmark) with default settings. Results were filtered for log2fold changes higher or lower than 2 or -2, *p*-values lower than 0.05, and a false discovery rate of 0.01 (Bonferroni). For differential expression analyses transcriptomic data was compared to *Ca*. Scheffleriplasma hospitalis genome, only.

Figure 2 summarising the metabolomic abilities of *Ca*. Micrarchaeum harzensis was prepared using templates from MetaboMAPS[69].

**Lipid analyses.** For the analysis of membrane lipids, cells were derived from 20 mL pure and co-culture in the exponential growth phase. Samples were taken for monitoring of cells via CARD-FISH prior to filtering (Supplementary Fig. 5). Cells were filtered onto 0.3 µm pore size 47 mm diameter glass fibre filters (GF75, Advantec MFS, Inc, CA, USA). Total lipids were extracted from the freeze-dried glass-fibre filters using a modified Bligh and Dyer method[70], as described earlier[71]. The extracts were dried under nitrogen and split into two aliquots, one left untreated and another hydrolysed with 1.5 M HCl in methanol by reflux at 130 °C for 2 h to remove the headgroups from the archaeal intact polar lipids (IPL) and release the core lipids (CLs). The pH was adjusted to 7 by adding 2 M KOH/MeOH (1:1 v/v) and, after the addition of water to a final 1:1 (v/v) ratio of $H_2O$-MeOH, extracted three times with dichloromethane (DCM). The DCM fractions were collected and dried over sodium sulphate. The dried samples were dissolved in hexane–2-propanol (99:1, vol/vol) and filtered over a 0.45-µm polytetrafluoroethylene filter. The extracts after acid-hydrolysis contained the IPL-derived CLs plus CLs, while the non-hydrolysed extracts consisted only of the CLs. Extracts were analysed by UHPLC–atmospheric pressure chemical ionisation (APCI) MS for archaeal CLs, including archaeol (diether, $C_{20}$ isoprenoid chains) and glycerol dialkyl glycerol tetraether (GDGTs, tetraether, $C_{40}$ side chain), according to ref. [72], with some modifications. Briefly, the analysis was performed on an Agilent 1260 UHPLC coupled to a 6130 quadrupole MSD in selected ion monitoring (SIM) mode. This allowed the detection of GDGTs with 0 to 4 cyclopentane moieties,

crenarchaeol as well as archaeol. The separation was achieved on two UHPLC silica columns (BEH HILIC columns, 2.1 × 150 mm, 1.7 µm; Waters Chromatography Europe BV, Etten-Leur, Netherlands) in series, fitted with a 2.1 × 5-mm precolumn of the same material (Waters Chromatography Europe BV, Etten-Leur, Netherlands) and maintained at 30 ˚C. Archaeal CLs were eluted isocratically for 10 min with 10% B, followed by a linear gradient to 18% B in 20 min, then a linear gradient to 100% B in 20 min, where A is hexane and B is hexane:isopropanol (9:1). The flow rate was 0.2 mL/min. Total run time was 61 min with a 20 min re-equilibration. Source settings were identical to ref. [73]. The typical injection volume was 10 µl of a 1 mg/mL solution (weighted dried Bligh and Dyer extract dissolved in hexane:isopropanol (99:1, v/v ratio)). The m/z values of the protonated molecules of archaeol and GDGTs were monitored. GDGTs were quantified by adding a $C_{46}$ GTGT internal standard[74]. A response factor derived from an archaeol:GDGT-0 standard (1:1) was used to correct for the difference in ionisation between archaeol and GDGTs.

The Bligh and Dyer extract (non-hydrolysed) and the acid-hydrolysed Bligh and Dyer extract were also analysed using ultra-high-performance liquid chromatography coupled to positive ion atmospheric pressure chemical ionisation/Time-of-Flight mass spectrometry (UHPLC-APCI/ToFMS) on an Agilent 1290 Infinity II UHPLC, equipped with an automatic injector, coupled to a 6230 Agilent TOF MS and Mass Hunter software. This additional analysis was performed to detect other archaeal lipids that were not included in the SIM method on the 6130 quadrupole MSD mentioned above. Separation of the archaeal lipids was achieved according to ref. [72] with some modifications using two silica BEH HILIC columns in series (2.1 × 150 mm, 1.7 µm; Waters Chromatography Europe BV, Etten-Leur, Netherlands) at a temperature of 25 °C. The injection volume was 10 µL. Compounds were isocratically eluted with 90% A and 10% B for the first 10 min, followed by a gradient to 18% B in 15 min, a gradient to 30% B in 25 min, and a linear gradient to 100% B in 30 min. A = hexane and B = hexane/isopropanol (9:1, v/v) and the flow rate was 0.2 mL/min. The conditions for the APCI source were identical to ref. [73,72]. Also, the fragmentor was set at 300 V. The ToFMS was operated in extended dynamic range mode (2 GHz) with a scan rate of 2 Hz. We assessed archaeal lipid distributions by monitoring m/z 600–1400. Archaeal lipids were identified by searching within 10 ppm mass accuracy for relevant [M + H] + signals.

**Metabolome analysis.** A pure culture of *Ca*. Scheffleriplasma hospitalis and a co-culture of *Ca*. Micrarchaeum harzensis and *Ca*. Scheffleriplasma hospitalis were inoculated as described above for 42 (pure culture) or 49 days (co-culture) until all available ferric iron was reduced and stationary phase was reached. Experiments were performed in triplicates. For metabolomic examination, 1 mL culture was sampled and stored at −80 °C until further analyses. Samples were taken every 7 days, along with samples for ferrous iron quantification to estimate growth, and samples for DNA extraction and CARD-FISH for further detection of *Ca*. Micrarchaeum harzensis and *Ca*. Scheffleriplasma hospitalis. Due to the low pH of the culture medium, 500 µL samples were amended by inducing sulphur precipitation through the addition of a spatula tip of $CaCO_3$ to each sample, mixing for 5 min at 2,000 rpm at room temperature. This treatment also led to cell lysis so that the analysis included also intracellular metabolites. After centrifugation for 5 min at 17,000 × *g* at room temperature, 50 µL of the supernatant was transferred to glass vials and dried under vacuum at 4 °C. Dried samples were stored at −80 °C until further analysis.

Online metabolite derivatization was performed using a Gerstel MPS2 autosampler (Muehlheim, Germany). Dried metabolites were dissolved in 15 µL of 2% methoxyamine hydrochloride in pyridine at 40 °C under shaking. After 90 min, an equal volume of N-Methyl-N-(trimethylsilyl)trifluoroacetamide (MSTFA) was added and held for 30 min at 40 °C. One µL of the sample was injected into an SSL injector at 270 °C in splitless mode. Gas chromatography/mass spectrometry (GC/MS) analysis was performed using an Agilent 7890 A GC equipped with a 30-m DB-35MS # 5-m Duraguard capillary column. Helium was used as carrier gas at a flow rate of 1.0 mL/min. The GC oven temperature was held at 100 °C for 2 min and increased to 300 °C at 10 K/min. After 3 min, the temperature was increased to 325 °C. The GC was connected to an Agilent 5975 C inert XL MSD, operating under electron ionisation at 70 eV. The MS source was held at 230 °C and the quadrupole at 150 °C. The total run time of one sample was 60 min. All GC/MS chromatograms were processed by using the MetaboliteDetector software[75].

**Structural analysis by electron microscopy.** For freeze-etching, both a pure culture of *Ca*. Scheffleriplasma hospitalis and a co-culture of *Ca*. Micrarchaeum harzensis and *Ca*. Scheffleriplasma hospitalis were concentrated by centrifugation (3000 × *g*). The concentrated cell pellet (1.5 µL) was applied onto a gold carrier, frozen in liquid nitrogen, and transferred into a freeze-etching device (CFE-50, Cressington, Watford, UK; *p* < 10⁻⁵ mbar). At T = 176 K, samples were fractured using a cold knife (T = 90 K); after sublimation of about 400 nm of surface water, the samples were shadowed with Platinum-Carbon at an angle of 45 degrees (1.5 nm), and an additional layer of pure Carbon (about 15 nm; both by electron-beam evaporation). Replicas were cleaned for 15 h on 70% $H_2SO_4$, washed three times in bidistilled water, taken up on 700 mesh (hex) grids and air-dried. For electron microscopy analysis at 200 kV, a transmission electron microscope JEM-

2100F (JEOL GmbH, Freising, Germany), equipped with a F416 CMOS camera (TVIPS, Gauting, Germany) under control of SerialEM v. 3.8[76] was used.

**Cryo-EM sample preparation.** For cryo-EM grid preparation, 100 µL of anaerobically grown culture was removed aseptically from the growth flask. Then, 2.5 µL of the culture was immediately applied to a freshly glow-discharged Quantifoil R2/2 Cu/Rh 200 mesh or R3.5/1 Au 200 mesh grid, adsorbed for 10 s, blotted for 4–5 s and plunge-frozen into liquid ethane using a Vitrobot Mark IV (Thermo Fisher Scientific Inc., Waltham, Massachusetts, USA), while the blotting chamber was maintained at 100% humidity at 10 °C. For half of the prepared grids 10 nm protein-A gold (CMC Utrecht, Netherlands) was pelleted by centrifugation (100,000 × g, 1 h, 4 °C) resuspended in 1:10 dilution of the growth medium and additionally added to the samples immediately prior to grid preparation.

**Cryo-ET data collection and analysis.** For tomographic data collection, the SerialEM software[76] was used on a Titan Krios G3 microscope using the Quantum energy filter (slit width 20 eV) and the K3 direct electron detector running in counting mode. Tilt series (46 in total) were collected in two sessions with a defocus range of −5 to −12 µm, collected between ±60° in a grouped dose symmetric scheme[77] with a 2° tilt increment. A total dose of 100 or 177 e−/Å$^2$ was applied over the entire series, and image data were sampled at a pixel size of 3.468 Å with 4 fractioned frames per tilt image. Unaligned tilt-movies frames were motion-corrected in IMOD and tilt series alignment using gold fiducials was performed in IMOD[78]. Tilt-series alignment without gold fiducials was performed using 250 ×250 pixel$^2$ patches with a fractional overlap of 33% within IMOD. Contrast transfer functions (CTFs) of the motion-corrected tilt series were estimated using CTFFIND4[79] and tomographic reconstruction of CTF-corrected aligned tilt-series was carried out using the SIRT algorithm implemented within Tomo3D[80,81]. Figure panels containing cryo-ET images were prepared using IMOD and Fiji[82]. Quantification of dividing *Ca*. Micrarchaeum harzensis cells was performed using medium magnification cryo-EM image data collected at a pixel size of 3.998 nm. Dividing cells were classified into associated or un-associated, whether they were in direct contact to the larger host cell *Ca*. Scheffleriplasma hospitalis, or separated, respectively.

**Reporting summary.** Further information on research design is available in the Nature Research Reporting Summary linked to this article.

## Data availability

The genome sequences generated in this study including annotations have been deposited at NCBI Genbank under Accession Numbers CP060530 and CP060531. Raw reads of transcriptomic data generated in this study are available as SRA files under BioSample accession codes SAMN15702898 and SAMN15702859. All generated raw files of phylogenetic trees and generated MS raw data of the lipid analysis can be found in the Zenodo repository [https://zenodo.org/record/4725435]. The MS raw data for metabolome analysis can be found in the *Digitale Bibliothek Braunschweig* repository [https://doi.org/10.24355/dbbs.084-202202151452-0]. Reference genomes and primers used in this study are provided in the Supplementary Information. Source data are provided with this paper.

## Code availability

All custom scripts used are available at Zenodo [https://zenodo.org/record/3839790][7].

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

## Acknowledgements

The authors thank Carola Berg for excellent technical assistance. T.A.M.B. is a recipient of a Sir Henry Dale Fellowship, jointly funded by the Wellcome Trust and the Royal Society (202231/Z/16/Z). T.A.M.B. would like to thank the Vallee Research Foundation, the European Molecular Biology Organization, the Leverhulme Trust and the Lister Institute for Preventative Medicine for support. A.S. was supported by the European Research Council (ERC STG ASymbEL: 947317), the Swedish Research Council (VR starting grant 2016-03559 to A.S.) and the NWO-I foundation of the Netherlands Organisation for Scientific Research (WISE fellowship). N.D. was supported by the NWO-I foundation of the Netherlands Organisation for Scientific Research (WISE fellowship to A.S.). L.V. was supported by the Soehngen Institute of Anaerobic Microbiology (SIAM) Gravitation grant (024.002.002) of the Netherlands Ministry of Education, Culture and Science (OCW) and the Netherlands Organization for Scientific Research (NWO). We thank Marianne Baas for technical support with the lipid analysis. S.K. and S.G. were partly funded by the German Research Foundation (DFG) (grant number: 252014092).

## Author contributions

S.K. and S.G. isolated the organisms, performed growth experiments, interpreted metagenomic and metatranscriptomic data and conducted together with T.R.N. and U.K. the lectin staining experiments. S.K., S.G. and J.G. contributed to the writing of the manuscript. J.G. designed the scientific study. T.R.N. and U.K. contributed to the writing of the manuscript. B.B. and C.P. performed PacBio and Illumina sequencing and were involved in bioinformatic analysis of the data. A.V.K. and T.A.M.B. performed cryo-ET analysis, interpretation of the respective data and contributed to the writing of the manuscript. R.R. conducted together with S.G. the electron microscopic analysis, interpreted the data and was involved in writing the manuscript. C.R. performed and interpreted the metabolome analysis and contributed to the writing of the manuscript. K.S.H. and K.H. contributed to metabolome analysis and writing of the manuscript. A.S. and N.D. analysed genome data, constructed phylogenetic trees and contributed to the writing of the manuscript. L.V. performed and interpreted the lipid analysis and contributed to the writing of the manuscript.

## Funding

## Competing interests

The authors declare no competing interests.
