## [Peer Review File · Nature Communications]

Reviewers' Comments:

Reviewer #1:

Remarks to the Author:

The manuscript describes the isolation of the first member of the Micrarchaeota in association with its host, a member of the Thermoplasmatales. Genomic, transcriptomic and metabolomic analyses were conducted to begin understanding the biology of these organisms, at least under laboratory conditions.

This is an important study as it expands the laboratory cultivation of members of the "DPANN". The authors conducted a variety of experiments and analyses but not all are very conclusive and integrated, leading in my opinion to some stretched conclusions. What exactly are the "critical growth factors" claimed by the title? Clearly the micrarchaeon has a larger genome for example than Nanoarchaeota but lacks pathways that would enable it to live independently. The authors provide metabolomic evidence for acquisition of some cellular precursors from its "host", as other studies have shown for example in *N. equitans* (Jahn et al 2004, Hamerly et al 2014). But there are important questions that remain un-answered and I was surprised that the combination of the several omics was not better connected with the microbiological characterization of the cultures. I find the FISH images and the lectin staining difficult to interpret, what is noise, what are cells in there. Quantitative PCR has been used to look at cell ratios, but is the host negatively impacted by the micrarchaeon? It is claimed that the dynamics of the association is different than that for example of *N. equitans*, but there are multiple studies that have shown that the number of free nanoarchaea depends on the stage of the culture and it is important to match those stages with the omic analyses (e.g. Giannone et al ISME J 2014). It doesn't appear here that the metabolomic time points were matched with the transcriptomes? Finding micrarchaea cells free in the medium, even some that have not completed cell division, doesn't necessarily mean they can go through cell division independently of host contact. If host cells lysed, all its ectobionts would be released.

An important finding was that the micrarchaeon grows better when the host forms a biofilm, maybe that was expected since it provides a structured environment where it can contact multiple host cells at the same time or switch between them? Live-dead staining to see impact on host cells (like in Jahn U. et al 2008) or EM of such biofilms would have been useful.

Minor comment: The source of the various lectins should be disclosed. It would also be useful in the lectin table to indicate which ones were effective in labeling each of the two organisms.

Reviewer #2:

Remarks to the Author:

The manuscript "Unraveling the critical growth factors for stable cultivation of (nano-sized_ Micrarchaeota" deals with the omics analyses of the co-culture *Ca. Scheffleriplasma hospitalis* (the host) and *Ca. Micrarchaeum harzensis*. The authors describe the first stable co-culture of a Micrarchaeum with a host, which apparently is free of any other organisms. Their detailed omics analysis suggests that Micrarchaeota rely on molecular building blocks for their host organism to make a living and on direct cell-cell contacts.

While I do find the omics analysis of the co-culture quite intriguing, I think that the results presented are lacking some novelty. The actual novelty of the authors' research is not presented and the study also lacks information for accurately reproducing the results. In detail, I have the following major concerns about the study:

- The authors provide information on the first stable co-culture of a member of the Micrarchaeota and their host. However, the co-culture is not publicly available and thus the results are not reproducible. Making cultures publicly available is quite important and with someone from the German culture collection DSMZ on the author list, the reader hoped that this has been taken care of. Other labs have also deposited co-cultures of Nanoarchaeota at culture collections to make them publicly available – please deposit the co-culture at a public culture collection (e.g. ATCC).
- The authors do not pay credit to previous studies that have heavily explored Micrarchaeota. For instance, the cell-cell interaction has been well documented based on cryo-EM from environmental studies (Comolli et al ISME) and so has the metabolic interaction of Micrarchaeota and their host.

A detailed comparison with these studies (what is the novelty here?) needs to be conducted.

- The title contains the degree of novelty that someone would expect in a paper published in Nature Communications, however, the content of the paper cannot live up to the title: a) "Unraveling the critical growth factors for stable cultivation" has not been addressed throughout the manuscript, although this would be the real novelty. Besides biofilm formation, which parameters are important for stable cultivation? b) How can this be performed for other nano-sized Micrarchaeota as the title refers to the entire phylum? These are questions, whose answers cannot be found in the manuscript but are crucial because this would be the real novelty. Just changing the title would not solve this issue either as the paper lacks a good physiological characterization of the micrarchaeota co-culture. Instead, the study is mainly based on omics resulting in knowledge that has previously been gathered from environmental studies and co-cultures of Nanoarchaeum equitans and its host. I've been following the field of nano-sized archaea for several decades but couldn't find any information that I had not heard of before from other studies in this paper. So, the novelty is unfortunately missing, which could be easily achieved by a good physiological characterization of the co-culture (growth curves, substrate tests, etc.).
- The authors name the two organisms with candidatus names, which is of course 100% adequate; one name is chosen based on the scientific contributions of a colleague. The naming is the first paragraph of the results and discussion. However, the names are not used; instead, the reader is left with acronyms. Why is this? The essential part of naming a microbe is to have a name that people can refer to. If you introduce a new name but use an acronym instead, a new name doesn't make sense at all. Please use the respective names throughout the manuscript also to give your colleague his due!
- It remains unclear how this study actually differs from the previous study also published by these authors on the exact same co-culture (though still contaminated with other organisms), particularly regarding the genomics part: Sci Rep 7, Article number: 3289. The paper in Sci Rep also contains the genomes and the transcriptomes. The genomic potential and the interactions based on transcriptomes should be presented in a cell cartoon to visualize how the metabolic interactions take place.
- The EM images are great and very similar to N. equitans but the research has gone beyond this. Please see Comolli et al who described the interaction of Micrarchaeota with the host based on cryo-EM for a much more detailed analysis. Doing cryo-EM and coupling this to the metabolic interactions could really result in a deeper understanding of the interaction of Micrarchaeota and their host and would present a real novelty (particularly paired with Transcriptomics when compared with and without symbiont). Reference study: The ISME Journal volume 3, pages 159–167 (2009)

Minor comments:

L64: "of" instead of "on"

L86: "aiming at understanding"

L100: Please do use the new names throughout the manuscript otherwise it's not an honor.

L 103: "at obtaining"

L110: Please do show the data. "data not shown" is inadequate for NPG.

L110-116: How has the purity of the culture been confirmed methodologically? I'm missing some standard 16S rRNA gene barcoding or qPCR on bacterial 16S rRNA genes; same for Eukarya. FISH targeting Archaea alone doesn't suffice here as others are not targeted.

L110-116: How did you make sure that there is only one of the originally two Thermoplasmatales in the culture? (shotgun sequencing will likely not be deep enough to verify absence of organisms)

L118: This part does need some sort of visualization, see major comment above.

L161: Since you performed multiple tests to identify genes that are differentially expressed, a p-value of 0.05 is not appropriate. Please perform a false discovery correcting, which will likely display a completely different picture of the differences in the transcriptome. The rest of the analysis needs to be adjusted accordingly.

L190: Same as above: p-value of 0.01 is not adequate in case of multiple testing.

Fig2: MVA pathway IV says "Archaea" in brackets but pathway II and III are archaeal, too. Please correct.

L550: Has quenching of the metabolites been performed prior to analysis? Quenching is pivotal for identifying volatile metabolites particularly at the low pH of the co-culture.

First of all, we want to thank all reviewers for their time and constructive instructions for improvement!

Reviewer #1 (Remarks to the Author):

The manuscript describes the isolation of the first member of the Micrarchaeota in association with its host, a member of the Thermoplasmatales. Genomic, transcriptomic and metabolomic analyses were conducted to begin understanding the biology of these organisms, at least under laboratory conditions.

This is an important study as it expands the laboratory cultivation of members of the "DPANN". The authors conducted a variety of experiments and analyses but not all are very conclusive and integrated, leading in my opinion to some stretched conclusions.

What exactly are the "critical growth factors" claimed by the title? Clearly the micrarchaeon has a larger genome for example than Nanoarchaeota but lacks pathways that would enable it to live independently. The authors provide metabolomic evidence for acquisition of some cellular precursors from its "host", as other studies have shown for example in *N. equitans* (Jahn et al 2004, Hamerly et al 2014). But there are important questions that remain un-answered and I was surprised that the combination of the several omics was not better connected with the microbiological characterization of the cultures.

We thank the reviewer for this feedback! We understand the feedback that the title is not really fitting, since the growth factor is not the main focus of this manuscript. We changed the title to "The importance of biofilm formation for cultivation of a Micrarchaeon and insights into interactions with its Thermoplasmatales host" and hope to better summarize our results in this way. Nevertheless, we are convinced that our manuscript possesses a crucial value for the field of DPANN archaea and archaeal symbionts. The manuscript reports on the isolation of a novel archaeal symbiont and through a multi-omic approach reveals insights into its interaction with its host, providing insights into the enigmatic life style of a diverse group of archaea. Also, we provide in this manuscript a blueprint for growth conditions that lead to stable growth of the co-culture (previous research revealed that a Micrarchaeon can be easily lost from co- and enrichment cultures; please see Golyshina *et al.*, 2017). Last but not least, we put a lot of effort over the last months in analysing the physical interaction points of the two organisms using cryo-electron tomography and were very successful in this respect.

I find the FISH images and the lectin staining difficult to interpret, what is noise, what are cells in there.

We see this point but are unfortunately not able to raise the resolution of the images. Using our available infrastructure, we had to choose between imaging individual cells, which leads to very strong signals in the biofilm flocks consisting of multiple cells behind each other and showing only biofilm flocks which would mask the low intensity signals of individual cells. I hope that the reviewer can accept this argumentation.

Quantitative PCR has been used to look at cell ratios, but is the host negatively impacted by the micrarchaeon?

The section "Characterization of *Ca. Micrarchaeum harzensis*-*Ca. Scheffleri*plasma *hospitalis* co-culture" was added to the revised manuscript, which addresses this topic. The host does not seem to be negatively impacted.

It is claimed that the dynamics of the association is different than that for example of *N. equitans*, but there are multiple studies that have shown that the number of free nanoarchaea depends on the stage of the culture and it is important to match those stages with the omic analyses (e.g. Giannone et al ISME J 2014).

Thank you for raising this point! Giannone *et al.*, 2014 and Jahn *et al.*, 2008 provide growth data for the *N. equitans*-*I. hospitalis* co-culture and the number of occupied *I. hospitalis* cells increases from exponential phase (30% occupation) to stationary phase (80% occupation). Our argumentation is based on the fact that cultivation of the co-culture is strongly dependent on growth in biofilm flocks, which was not reported for the *N. equitans*-*I. hospitalis* interaction. We revised the manuscript for clarification.

It doesn't appear here that the metabolomic time points were matched with the transcriptomes?

The metabolic and transcriptomic analyses are independent experiments. To better compare them, we highlighted the time point of sampling for transcriptomic analysis in the growth curves of the metabolic analyses (Supplemental Figure S4).

Finding micrarchaea cells free in the medium, even some that have not completed cell division, doesn't necessarily mean they can go through cell division independently of host contact. If host cells lysed, all its ectobionts would be released.

We thank the reviewer for bringing up this point and addressed this topic in an additional section showing cryo-ET micrographs. A quantitative analysis revealed that the number of dividing Micrarchaeota attached to the host is higher than the number of dividing Micrarchaeota without direct contact and we revised this section accordingly (Figure 8).

An important finding was that the micrarchaeon grows better when the host forms a biofilm, maybe that was expected since it provides a structured environment where it can contact multiple host cells at the same time or switch between them? Live-dead staining to see impact on host cells (like in Jahn U. et al 2008) or EM of such biofilms would have been useful.

Live-Dead staining was conducted. Biofilm flocks seem to consist of dead cells in the centre of the flocks and living cells at the surface. Still, we were not able to distinguish in this analysis between *Ca. Micrarchaeum harzensis* and *Ca. Scheffleri*plasma *hospitalis* cells. Nevertheless, the growth data that was added to the manuscript shows at least no obvious negative impact on the growth of the host.

Minor comment: The source of the various lectins should be disclosed. It would also be useful in the lectin table to indicate which ones were effective in labeling each of the two organisms.

Corrected.

Reviewer #2 (Remarks to the Author):

The manuscript “Unraveling the critical growth factors for stable cultivation of (nano-sized_ Micrarchaeota” deals with the omics analyses of the co-culture *Ca. Schefflerioplasma hospitalis* (the host) and *Ca. Micrarchaeum harzensis*. The authors describe the first stable co-culture of a *Micrarchaeum* with a host, which apparently is free of any other organisms. Their detailed omics analysis suggests that *Micrarchaeota* rely on molecular building blocks for their host organism to make a living and on direct cell-cell contacts.

While I do find the omics analysis of the co-culture quite intriguing, I think that the results presented are lacking some novelty. The actual novelty of the authors’ research is not presented and the study also lacks information for accurately reproducing the results. In detail, I have the following major concerns about the study:

- The authors provide information on the first stable co-culture of a member of the *Micrarchaeota* and their host. However, the co-culture is not publicly available and thus the results are not reproducible. Making cultures publicly available is quite important and with someone from the German culture collection DSMZ on the author list, the reader hoped that this has been taken care of. Other labs have also deposited co-cultures of *Nanoarchaeota* at culture collections to make them publicly available – please deposit the co-culture at a public culture collection (e.g. ATCC).

Unfortunately, it is currently not possible to submit the organisms to a culture collection such as DSMZ. In particular, the problem is that these organisms/co-cultures cannot be stored as cryo-stocks and need to be continuously grown to be maintained. This is even more challenging considering the doubling time of 7 days for these co-cultures. In turn, the strain collections are not able to maintain those cultures and ensure the presence of the *Micrarchaeon* which can easily be lost during continuous cultivation. However, we are always happy to provide cultures to anybody requesting them. The pure culture of *Ca. Schefflerioplasma hospitalis* is deposited at Japan Collection of Microorganisms (JCM accession number: JCM 39074). This information was added to the manuscript (lines 433-434).

- The authors do not pay credit to previous studies that have heavily explored *Micrarchaeota*. For instance, the cell-cell interaction has been well documented based on cryo-EM from environmental studies (Comolli et al ISME) and so has the metabolic interaction of *Micrarchaeota* and their host. A detailed comparison with these studies (what is the novelty here?) needs to be conducted.

We apologise for this oversight and have corrected this. Please note that the comparison with cryo-EM from environmental studies (Comolli *et al.*, 2009) is part of another manuscript of ours (Gfrerer *et al.*, 2021; <https://doi.org/10.1101/2021.04.28.441871>), which mentions among others the advantages of a culture with defined composition compared to environmental samples.

- The title contains the degree of novelty that someone would expect in a paper published in Nature Communications, however, the content of the paper cannot live up to the title: a) “Unraveling the critical growth factors for stable cultivation” has not been addressed throughout the manuscript, although this would be the real novelty. Besides biofilm formation, which parameters are important for stable cultivation? b) How can this be performed for other nano-sized *Micrarchaeota* as the title refers to the entire phylum? These are questions, whose answers cannot be found in the manuscript but are crucial because this would be the real novelty. Just changing the title would not solve this issue either as the paper lacks a good physiological characterization of the *micrarchaeota* co-culture. Instead, the study is mainly based on omics resulting in knowledge that has previously been gathered from environmental studies and co-cultures of *Nanoarchaeum equitans* and its host. I’ve been following the field of nano-sized archaea for several decades but couldn’t find any information

that I had not heard of before from other studies in this paper. So, the novelty is unfortunately missing, which could be easily achieved by a good physiological characterization of the co-culture (growth curves, substrate tests, etc.).

A section with the physiological characterisation of the cultures was added, also the title was changed to “The importance of biofilm formation for cultivation of a Micrarchaeon and insights into interactions with its Thermoplasmatales host”.

This study brings a new factor to the field, as it reports on the first isolation of a Micrarchaeon in a co-culture with its host following a characterization of this culture. Previous studies on Micrarchaeota were carried out on environmental studies. This manuscript describes analyses under controlled conditions without contaminants. Comparable studies are published for only one DPANN member so far; *Nanoarchaeum equitans*. Even though *N. equitans* represents the closest characterized relative, similar types of symbiosis for Nanoarchaeota and Micrarchaeota are not to be expected, since these are evolutionary distant phyla in the diverse group of DPANN archaea (Dombrowski *et al.*, 2020; <https://doi.org/10.1038/s41467-020-17408-w>). This work reports on the analyses of the third complete Micrarchaeum genome to date and while growth in biofilms was described earlier, we report on an enrichment strategy where we selected for biofilm formation of the host. The isolation of the host organism proved to be essential, since it allowed us to conduct comparative studies to elucidate the EPS and lipid composition of the Micrarchaeon. Besides biofilm formation, this study could show a direct physical contact between the Micrarchaeon cell and its host via electron microscopy studies, indicating a similar symbiosis like Nanoarchaeota (which was not expected as mentioned above). During the analysis of our micrographs, we were able to clearly identify the two organisms by cell morphology, which is not entirely possible for electron micrographs of environmental samples. Another novelty of this work would include the investigation of the transcriptomic response of *Ca. Scheffleriplasma hospitalis* to co-cultivation with the Micrarchaeon. Furthermore, the manuscript now also contains another section presenting cryo-ET data, that provides more in-depth analyses how the interaction is achieved and also shows a new form of ribosome arrangement in the Micrarchaeon.

- The authors name the two organisms with candidatus names, which is of course 100% adequate; one name is chosen based on the scientific contributions of a colleague. The naming is the first paragraph of the results and discussion. However, the names are not used; instead, the reader is left with acronyms. Why is this? The essential part of naming a microbe is to have a name that people can refer to. If you introduce a new name but use an acronym instead, a new name doesn't make sense at all. Please use the respective names throughout the manuscript also to give your colleague his due!

We apologize and corrected this according to the reviewer's suggestion.

- It remains unclear how this study actually differs from the previous study also published by these authors on the exact same co-culture (though still contaminated with other organisms), particularly regarding the genomics part: Sci Rep 7, Article number: 3289. The paper in Sci Rep also contains the genomes and the transcriptomes. The genomic potential and the interactions based on transcriptomes should be presented in a cell cartoon to visualize how the metabolic interactions take place.

Thank you for this comment. This paper is a significant extension of our previous work: First of all, we have been able to obtain a pure co-culture of the symbiont and host and obtained complete genomes for both organisms, which revealed that the Micrarchaeon has more metabolic functions

than previously assumed (e.g. with regard to carbohydrate metabolism). We furthermore provide a more in-depth characterisation of the physiological potential of the Micrarchaeon, which was crucial for the metabolomic analysis. We also provide insights into the transcriptomic response of the host organism to co-cultivation with the Micrarchaeon, which is evidently a new aspect for the investigation of the interaction between the two archaea and provide new insights into the cell biology and interactions using high-end microscopy (see below).

- The EM images are great and very similar to *N. equitans* but the research has gone beyond this. Please see Comolli et al who described the interaction of Micrarchaeota with the host based on cryo-EM for a much more detailed analysis. Doing cryo-EM and coupling this to the metabolic interactions could really result in a deeper understanding of the interaction of Micrarchaeota and their host and would present a real novelty (particularly paired with Transcriptomics when compared with and without symbiont). Reference study: The ISME Journal volume 3, pages 159–167 (2009)

We performed cryo-ET to characterise the interaction site of *Ca. Micrarchaeum harzensis* and *Ca. Scheffleriplasma hospitalis* in more detail. This analysis also revealed new information on the arrangement of ribosomes in the cytoplasm. The data was added to the manuscript.

Minor comments:

L64: “of” instead of “on”

Corrected.

L86: “aiming at understanding”

Corrected.

L100: Please do use the new names throughout the manuscript otherwise it’s not an honor.

Corrected.

L 103: “at obtaining”

Corrected.

L110: Please do show the data. “data not shown” is inadequate for NPG.

We changed the corresponding sentence in the manuscript. We conducted this step according to our observation of growth at lower temperatures in the laboratory but did not collect triplicate growth curves of the strain that would lead to a value for the growth optimum.

L110-116: How has the purity of the culture been confirmed methodologically? I’m missing some standard 16S rRNA gene barcoding or qPCR on bacterial 16S rRNA genes; same for Eukarya. FISH targeting Archaea alone doesn’t suffice here as others are not targeted.

The composition of cultures was verified via PCR with organism-specific primers, periodically. This information was added to the revised version of the manuscript, correspondingly (lines 115-118 and 124-125).

L110-116: How did you make sure that there is only one of the originally two Thermoplasmatales in the culture? (shotgun sequencing will likely not be deep enough to verify absence of organisms)

Besides verifying composition of cultures via PCR with organism-specific primers (see also comment above), Illumina and Pacbio sequencing of the culture for genomic analysis did not show any other organism beside *Ca. Micrarchaeum harzensis* and *Ca. Scheffleriplasma hospitalis*.

L118: This part does need some sort of visualization, see major comment above.

A new figure was added (Figure 2).

L161: Since you performed multiple tests to identify genes that are differentially expressed, a p-value of 0.05 is not appropriate. Please perform a false discovery correcting, which will likely display a completely different picture of the differences in the transcriptome. The rest of the analysis needs to be adjusted accordingly.

Corrected.

L190: Same as above: p-value of 0.01 is not adequate in case of multiple testing.

Corrected.

Fig2: MVA pathway IV says "Archaea" in brackets but pathway II and III are archaeal, too. Please correct.

Corrected.

L550: Has quenching of the metabolites been performed prior to analysis? Quenching is pivotal for identifying volatile metabolites particularly at the low pH of the co-culture.

No quenching was performed. The applied method is not suitable for the analysis of volatile metabolites in the supernatant, as we need to remove any residual water by freeze drying prior derivatization.

Reviewers' Comments:

Reviewer #1:

Remarks to the Author:

The revised manuscript satisfactorily addresses my original comments and requests

Reviewer #2:

Remarks to the Author:

The authors have carefully revised the manuscript and taken care of most of the comments that I raised.

The critical growth factors initially mentioned in the title of the paper were the most interesting part but the content of the original paper was low on that topic. Instead of determining on the critical growth factors which might help cultivating more DPANN, the authors have decided to change the title, which is unfortunate. I also have concerns regarding the novelty of the co-culture and the findings presented:

- Transcriptome and genome were published here <https://www.nature.com/articles/s41598-017-03315-6>

- ultrastructure of the DPANN was published a few days ago

<https://journals.asm.org/doi/abs/10.1128/AEM.01553-21>

- A recent paper already describes the first DPANN co-culture

<https://www.pnas.org/content/119/3/e2115449119>

The most exciting part for me was the arrangement of the ribosomes in the DPANN (this warrants further analyses I hope).

The techniques and methods have been applied flawlessly in my opinion and resulted in a great dataset. I do not have any further questions.

Answers to reviewer comments:

We would like to thank reviewer #1 and #2 for reevaluating the manuscript and the generally positive comments. Reviewer #2 still had some concerns that we would like to address here:

The reviewer has doubts regarding the question whether we were able to reveal critical growth factors for the enrichment of DPANN in this manuscript. We would like to draw the reviewer's attention to the determined necessity to grow the co-culture in the form of a biofilm as well as the necessity to limit growth of the host organism by adjusting growth conditions to be suboptimal for the host. In the future, we will be able to use this knowledge by isolating members of the DPANN via biofilm cultivation platforms. Continuous monitoring of biofilm growth and composition together with the possibility of multiparallel experimentation using microfluidic platforms will be extremely helpful for future studies. Of note first experiments in our group revealed that we can grow our co-culture also in microfluidic PDMS chip systems.

The reviewer has concerns regarding the novelty of the study and lists specifically three manuscripts.

The first manuscript by Krause *et al.* was published by our group. It contains transcriptomic and metagenomic data of an enrichment culture and not of the here presented co-culture. Moreover, we present here not only fully sequenced genomes but also the impact of co-culturing on the transcriptome of the host organism.

The next manuscript by Gfrerer *et al.* was also published by us. It describes that the two organisms of the enrichment culture can be discriminated by their surface structure. Hence, this publication does not limit the novelty of the here evaluated study but was key for setting up a lot of the here described experiments as we were now able to discriminate between the Micrarchaeon and its host using electron microscopy. Furthermore, the manuscript by Gfrerer *et al.* shows data about single cells only, while this manuscript reveals details about the interaction between the two organisms of the co-culture.

Last but not least, the reviewer mentions a very interesting study that was accepted while our manuscript was already under consideration by *nature communication*. The authors were able to isolate a thermoacidophilic co-culture of a Micrarchaeon with a member of the genus *Metallosphaera*. We refer to this manuscript also in the revised version of our study. It is interesting to observe that the physiological variability of Micrarchaeota members seems to be higher than expected. Contrary to our study, the co-culture enriched by Sakai *et al.* grows aerobically and was derived from spring water. We will try to trace the physiological differences back to genomic and transcriptomic variations in future studies.